# Chromatin organization in the female mouse brain fluctuates across the oestrous cycle

Ivana Jaric[1], Devin Rocks[1], John M. Greally [2], Masako Suzuki [2] & Marija Kundakovic [1]

Male and female brains differ significantly in both health and disease, and yet the female brain has been understudied. Sex-hormone fluctuations make the female brain particularly dynamic and are likely to confer female-specific risks for neuropsychiatric disorders. The molecular mechanisms underlying the dynamic nature of the female brain structure and function are unknown. Here we show that neuronal chromatin organization in the female ventral hippocampus of mouse fluctuates with the oestrous cycle. We find chromatin organizational changes associated with the transcriptional activity of genes important for neuronal function and behaviour. We link these chromatin dynamics to variation in anxiety-related behaviour and brain structure. Our findings implicate an immediate-early gene product, Egr1, as part of the mechanism mediating oestrous cycle-dependent chromatin and transcriptional changes. This study reveals extreme, sex-specific dynamism of the neuronal epigenome, and establishes a foundation for the development of sex-specific treatments for disorders such as anxiety and depression.

[1] Department of Biological Sciences, Fordham University, 441 E. Fordham Road, Bronx, NY 10458, USA. [2] Center for Epigenomics, Department of Genetics, Albert Einstein College of Medicine, 1301 Morris Park Avenue, Bronx, NY 10461, USA. Correspondence and requests for materials should be addressed to M.K. (email: mkundakovic@fordham.edu)

Brain structure and function are sexually dimorphic. As neuroscience research has largely focused on the male brain and behaviour[1–3], the female brain and, in particular, its inherent dynamics have been left underexplored. During the mammalian reproductive period, the female brain is exposed to fluctuating hormone levels over the cycles known as menstrual (in humans) or oestrous (in rodents). While being indispensable for the reproductive function, fluctuating sex hormone levels dynamically affect female brain morphology[4–7], function[8–10] and neurochemistry[11], and are likely contributors to female-specific risk for certain neuropsychiatric conditions.

Anxiety and depression are the most frequent psychiatric disorders, affecting ~20% of the world's population, and are two times more prevalent in women than in men[12,13]. Several lines of evidence indicate that sex-hormone fluctuations represent a major risk factor for the increased female vulnerability to these disorders. The female bias in depression rates is first noted upon the occurrence of menarche in girls, peaks at perimenopause characterised by extreme hormone fluctuations and finally drops when stable, low oestradiol levels are established at post-menopause[14]. In addition, the premenstrual dysphoric disorder, affecting 5–8% of women and requiring antidepressant treatment[15], strongly reflects an impact that natural hormonal shifts can have on female mood and well-being. It is, therefore, of great importance to understand the molecular mechanisms that underlie the sex hormone-induced, dynamic nature of the female brain.

We hypothesised that sex hormones dynamically impact female brain structure and function by epigenetic (transcriptional regulatory) mechanisms. Regulation of chromatin accessibility is a major regulatory mechanism controlling gene expression[16], and we suspected that chromatin reorganization may occur in response to cycling hormone levels. Global differences in gene expression have been found during the oestrous cycle in the prefrontal cortex[17] and the hippocampus[18]. While the oestrous cycle effects on the brain epigenome have not been explored, oestradiol is known to induce chromatin remodelling that drives transcriptional changes in dividing cells[19]. Studies in ovariectomised female mice have also suggested that the epigenomes of adult hippocampal cells are responsive to oestrogen[20].

Here, we present a rodent study performed under natural physiological conditions with systematic oestrous cycle determination in female mice and the inclusion of age-matched males. We characterise the effect of the oestrous cycle and sex on neuronal chromatin accessibility and gene expression in the ventral hippocampus in relation to behavioural and structural phenotypes. Our study reveals an extreme dynamism of the female neuronal epigenome, associated with changes in gene expression, synaptic plasticity and anxiety-related behaviour, providing a paradigm and mechanism for studying sex differences in brain disorders.

## Results

**Study design.** The ovarian cycle in females is initiated at puberty and ceases at menopause. In human females, each menstrual cycle typically lasts 28–35 days and includes the follicular (high oestrogen–low progesterone) phase and the luteal (low oestrogen–high progesterone) phase (Fig. 1a). Rodents represent a good model to study the effects of naturally occurring sex-hormone fluctuations on brain and behaviour. Although the rodent oestrous cycle differs by having four phases and by being shorter in duration (typically 4–5 days), the human follicular and luteal phases are mimicked by the rodent proestrus (high oestrogen–low progesterone) and early dioestrus (low oestrogen–high progesterone) phases, respectively (Fig. 1a). In

this study, we used female C57BL/6J mice, and to determine their cycling patterns, the oestrous cycle stage was checked by vaginal cytology daily, from 6 to 8 weeks of age (Methods; Supplementary Fig. 1a, b). We confirmed that vaginal cytology closely reflects oestradiol and progesterone levels in serum and the hippocampus (Supplementary Fig. 1c). At 8 weeks, females were assigned to dioestrus or proestrus groups and tested with aged-matched males (Fig. 1a). The oestrous cycle stage was re-confirmed after each behavioural test and at the time of tissue collection for molecular and histological studies.

**Anxiety-like behaviour is oestrous cycle- and sex-dependent.** We first examined the effect of the oestrous cycle and sex on anxiety-like behaviour (from 8 to 10 weeks of age) using three well-established tests: open-field, light–dark box and elevated-plus maze tests. Tests were separated by one oestrous cycle (or by 4–5 days in males), making sure that each female animal is always tested at its assigned oestrous cycle stage (dioestrus or proestrus). Across all tests, female mice in the dioestrus phase exhibited higher indices of anxiety-like behaviour compared to proestrus females and males (Fig. 1b). In the open field, we found a significant difference between groups ($F_{(2,39)} = 5.93$, $P = 0.006$; one-way ANOVA), with dioestrus females spending less time in the centre than proestrus females ($P = 0.006$; Tukey's post hoc test) and males ($P = 0.044$; Tukey's post hoc test). Similarly, there was a significant effect of the group in the light–dark box test ($F_{(2,37)} = 21.63$, $P < 0.001$; one-way ANOVA), where we found that dioestrus females spent less time in the light compartment compared to proestrus ($P < 0.001$; Tukey's post hoc test) and male ($P < 0.001$; Tukey's post hoc test) animals. Finally, the elevated-plus maze test revealed a significant group difference ($F_{(2,37)} = 5.33$, $P = 0.009$; one-way ANOVA): dioestrus females spent significantly less time in open arms compared to proestrus females ($P = 0.008$; Tukey's post hoc test), and there was a similar trend in the dioestrus-male comparison ($P = 0.08$; Tukey's post hoc test). No significant difference was found in any of the proestrus-male comparisons across all tests. We also did not find any significant difference in general activity levels among the three groups, including total distance travelled in the open-field ($F_{(2,39)} = 0.83$, $P = 0.44$; one-way ANOVA) and elevated-plus maze ($F_{(2,37)} = 0.09$, $P = 0.91$; one-way ANOVA) tests, showing that the oestrous cycle specifically impacted indices of anxiety-like behaviour. Overall, these findings imply that a physiological drop in oestrogen may increase the risk for anxiety in females, consistent with previous studies in rats[21] and with human data[8,9].

**Chromatin organization fluctuates across the oestrous cycle.** We next tested whether chromatin reorganization may occur during the oestrous cycle, reflecting a transcriptional regulatory mechanism responding to sex hormone levels and contributing to the observed behavioural differences. Gene regulatory regions with accessible, open chromatin structure allow the binding of transcription factors and gene activation, whereas closed chromatin configuration is not permissive for transcription[16]. Importantly, chromatin states are brain region-specific and cell-type-specific, with significant differences being reported between neuronal and non-neuronal cells[22]. We, therefore, assessed the effects of the oestrous cycle and sex on neuronal chromatin organization in the ventral hippocampus, an area strongly implicated in emotion regulation[23].

For this assessment, we performed the assay for transposase-accessible chromatin using sequencing (ATAC-seq)[24] on purified neuronal nuclei[22] isolated from the ventral hippocampus from proestrus, dioestrus and male animals (Fig. 1c). Purification of neuronal (NeuN+) nuclei was performed by fluorescence-

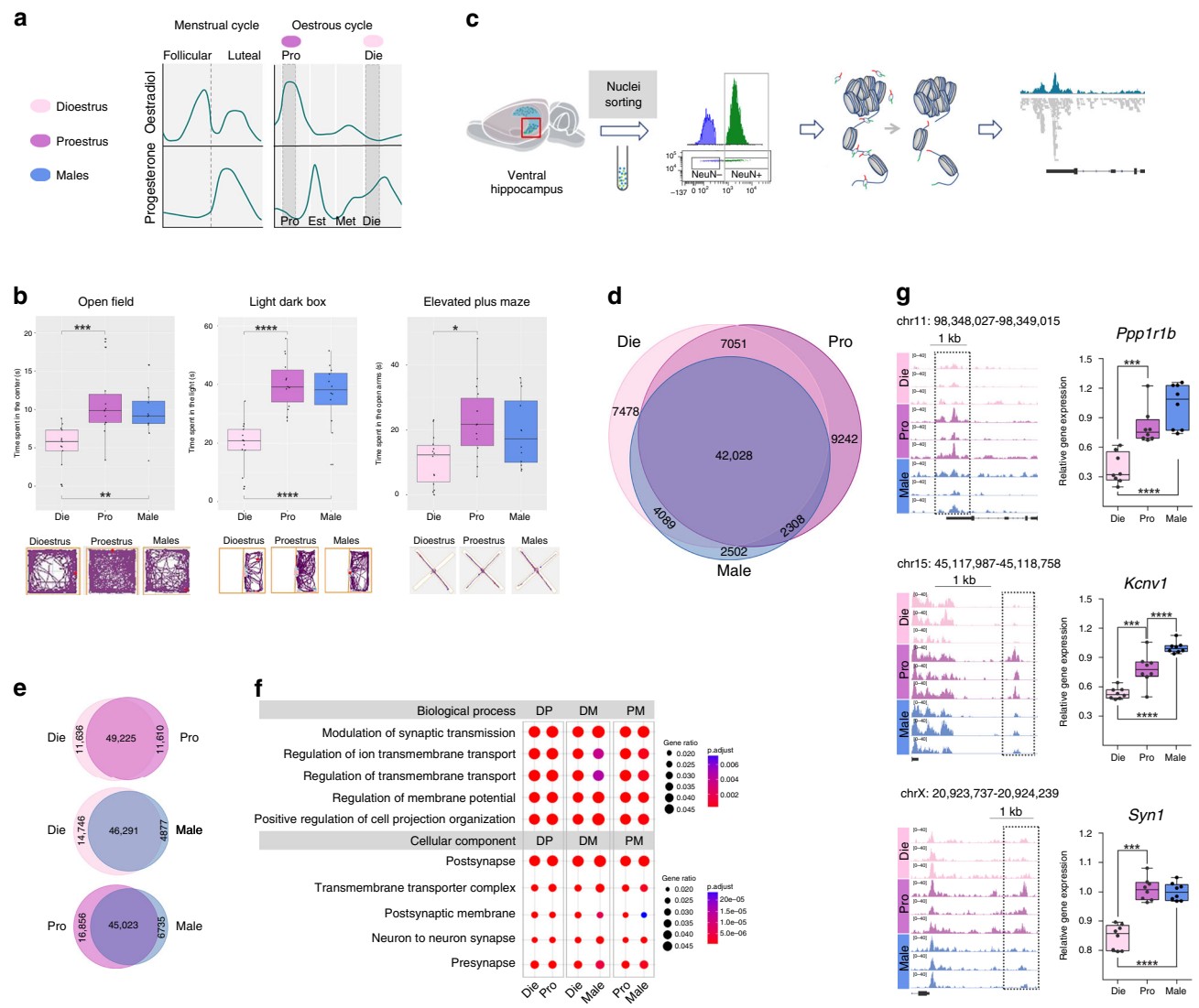

**Fig. 1** Anxiety-like behaviour and neuronal chromatin organization fluctuate across the oestrous cycle and differ between sexes. **a** Two stages of the oestrous cycle, proestrus (Pro, purple; high oestrogen–low progesterone) and early dioestrus (Die, light pink; low oestrogen–high progesterone) in mice mimic follicular and luteal phases in humans, respectively; Pro and Die females are tested with age-matched males (Male, blue) in all experiments. **b** Anxiety tests included: open field (time spent in the centre); light–dark box (time spent in the light), and elevated-plus maze (time spent in open arms). Shown are individual data (box plots, $n = 12$–16 animals/group) and representative track plots. **c** Scheme of the ATAC-seq assay of FACS-purified neuronal nuclei from the ventral hippocampus. **d** Venn diagrams show the number of open chromatin regions (ATAC-seq peaks) in each group and their overlap ($n = 3$ biological replicates/group). **e** Comparison of ATAC-seq peaks in Die vs. Pro, Die vs. Male, and Pro vs. Male. **f** GO analysis of genes showing differential chromatin accessibility between the groups: DP, Die-Pro comparison; DM, Die-Male comparison; PM, Pro-Male comparison. For each comparison, GO analysis was done twice: e.g. DP_Die represents genes with chromatin open in Die/closed in Pro; DP_Pro represents genes open in Pro/closed in Die; colours indicate adjusted $p$-values and dot size indicates gene ratio; top five GO terms for biological process and cellular component are shown. **g** Chromatin accessibility profiles of the *Ppp1r1b*, *Kcnv1* and *Syn1* promoters ($n = 3$ biological replicates/group) and corresponding mRNA levels ($n = 8$ animals/group). Shown are genomic coordinates of differential ATAC-seq peaks and their distance to the TSS. Box plots (box, 1st–3rd quartile; horizontal line, median; whiskers, min/max); $*p < 0.05$; $**p < 0.01$; $***p < 0.001$; $****p < 0.0001$ (one-way ANOVA with the Tukey's post hoc test)

activated nuclei sorting, using an antibody against the neuronal nuclear marker NeuN, and allowed a high-resolution, cell-type-specific chromatin accessibility profiling. The ATAC-seq assay is based on the activity of the enzyme Tn5 transposase, which cuts DNA and inserts sequencing adapters into open genomic regions, allowing the genome-wide identification of DNA elements located at sites of open chromatin. Following amplification, sequencing and bioinformatic analysis of the accessible DNA, each open chromatin region is detected as an enrichment signal or peak (Fig. 1c). We considered only high-confidence peaks which were detected in at least two out of three biological replicates per each group.

We found 50,000–60,000 open chromatin regions in each group—dioestrus, proestrus and males—with the majority (42,028, 69.3–82.5%) regions in common to all groups (Fig. 1d). A smaller number (7478 regions) was specific to dioestrus, a comparable number (9242 regions) specific to proestrus and only 2502 regions were specific to males (Fig. 1d). Remarkably, 32.1% of the regions showed differential chromatin accessibility between dioestrus and proestrus females (Fig. 1e; Supplementary Data 1), implying that hippocampal neuronal chromatin undergoes significant reorganization during the few days of the oestrous cycle. This within-female chromatin variability was found to be of a similar extent as between-sex variability, as we found that 29.8%

and 34.4% of the regions show differential chromatin accessibility in the dioestrus female versus male and proestrus female versus male comparisons, respectively (Fig. 1e; Supplementary Data 1).

**Chromatin dynamics affect neuronal function-relevant genes.** To examine the functional significance of the observed within- and between-sex variation in chromatin organization, we performed gene ontology analysis on the genes proximal to the loci with differential chromatin accessibility for each of the three comparisons: dioestrus versus proestrus (DP), dioestrus versus males (DM) and proestrus versus males (PM) (Fig. 1f). This analysis indicated that the genes regulated by the differential chromatin accessibility are involved in the regulation of synaptic transmission, membrane potential and neurite organization, among others, indicating that the observed chromatin differences play an important role in the regulation of neuronal function (Fig. 1f; Supplementary Data 2). Among candidate genes, *Ppp1r1b* (encoding Darpp32) was selected based on its significance for normal brain function and its role in psychiatric disorders, such as schizophrenia and anxiety[25,26] (Fig. 1g). We also found that mRNA expression of this gene varies across the three examined groups ($F_{(2,21)} = 23.3$, $P < 0.001$; one-way ANOVA), suggesting oestrous cycle- and sex-specific gene regulation. Around the *Ppp1r1b* transcription start site (TSS), we found the chromatin to be open in proestrus females and closed in dioestrus females, which was consistent with the higher expression of *Ppp1r1b* in proestrus compared to dioestrus females (Fig. 1g, $P = 0.001$; Tukey's post hoc test). We found that *Ppp1r1b* mRNA levels were also higher in males than in dioestrus females (Fig. 1g, $P < 0.001$; Tukey's post hoc test), and that this difference was associated with increased chromatin accessibility in males within a region ~6 kb upstream of the TSS, revealing sex-specific chromatin regulation at this gene (Supplementary Data 1, Supplementary Fig. 2). In addition, genes important for neuronal excitability and synaptic function, *Kcnv1* (encoding the Potassium voltage-gated channel modifier subfamily V member 1) and *Syn1* (encoding a synaptic vesicle protein Synapsin I), respectively, showed differential chromatin accessibility between groups (Fig. 1g). These chromatin differences corresponded to differences in gene expression (*Kcnv1*, $F_{(2,21)} = 40.59$, $P < 0.001$; *Syn1*, $F_{(2,21)} = 47.55$, $P < 0.001$; one-way ANOVA), further confirming functional significance of the observed chromatin organizational changes (Fig. 1g).

**Chromatin changes are enriched in serotonergic synapse genes.** We then went on to explore the functional pathways that could be contributing to behavioural differences. For each comparison—dioestrus versus proestrus (DP), dioestrus versus males (DM) and proestrus versus males (PM)—we merged all genes that show differential chromatin accessibility (open or closed) between the two compared groups (Fig. 2a). Many important pathways, including the major neurotransmitter systems and signalling pathways, showed significant enrichment in each of the comparisons (Fig. 2a; Supplementary Data 3). However, there were only two pathways—Serotonergic synapse and Hedgehog signalling—that showed an enrichment in DP and DM comparisons, but not in the PM comparison (Fig. 2a), which are the groups with differences in the anxiety tests (Fig. 1b). Since serotonergic transmission has been strongly implicated in anxiety and is targeted by the major anti-anxiety and antidepressant drugs[27], as well as by oestrogen[28], we decided to explore further the candidate genes belonging to this pathway. Among the total of 132 genes within the mouse KEGG serotonergic synapse pathway (mmu04726), 52 genes showed differential local chromatin accessibility in both DP and DM comparisons, with these genes

encoding receptors, signalling molecules, ion channels and enzymes important for serotonergic transmission (Fig. 2b). Three candidate genes implicated in anxiety and depression were selected: *Htr2b*[29] (encoding the Serotonin receptor 2b), *Cacna1c*[30] (encoding the α1C subunit of the voltage-gated L-type calcium channel Ca$_v$1.2) and *Ptgs1* (encoding a prostaglandin-synthesising enzyme Cyclooxygenase-1). We were able to show that differential chromatin organization in the putative regulatory regions of these genes corresponded to differential gene expression among the three groups (Fig. 2c; *Hrt2b*, $F_{(2,21)} = 12.87$, $P < 0.001$; *Cacna1c*, $F_{(2,21)} = 115.07$, $P < 0.001$; *Ptgs1*, $F_{(2,21)} = 17.58$, $P < 0.001$; one-way ANOVA). While *Htr2b* and *Ptgs1* exhibited more accessible chromatin and higher mRNA expression in dioestrus compared to both proestrus and males, *Cacna1c* exhibited a more complex oestrous cycle- and sex-specific pattern of chromatin organization and gene expression (Fig. 2c). Our data suggest that differential chromatin organization within serotonergic genes provides a plausible molecular mechanism for sex hormone- and sex-dependent variation in serotonergic signalling[28] and related behaviours, including anxiety-like behaviour.

**Proestrus is associated with opening of Egr1 binding sites.** Each stage of the oestrous cycle has a complex hormonal profile, including varying levels of oestradiol and progesterone (Fig. 1a). Previous studies have shown that oestradiol plays a major role in the induction of dendritic spine growth[5] and in the low-anxiety phenotype[31] within the proestrus phase. Oestradiol can function through both nuclear and membrane receptors, and the membrane-associated oestrogen signalling is implicated in rapid oestrogen effects in the hippocampus[8]. To explore possible mediators of oestrogen-induced changes in chromatin organization, we examined the DNA sequence characteristics of the loci open in proestrus (high-oestrogenic phase) and closed in dioestrus (low-oestrogenic phase, Supplementary Data 1). Enrichment of oestrogen response elements would be consistent with the classical, nuclear receptor-mediated mechanism, whereas over-representation of motifs for other transcription factors could point towards a membrane receptor-dependent mechanism (Supplementary Fig. 3). The Homer motif analysis identified five proestrus-specific motifs in the proestrus–dioestrus comparison—motifs for the Mef2 family members (Mef2a–d) and Egr1 (Fig. 3a; Supplementary Data 4)—all of which have been associated with kinase pathways[32,33], implying the involvement of a membrane oestrogen receptor (ER; Supplementary Fig. 3). Consistent with this model, we found that anxiety-related ERβ[34] is primarily localised in the cell membrane and cytosol, and co-expressed with our candidate protein Darpp32 in ventral hippocampal neurons (Supplementary Fig. 4).

Among the Homer-identified transcription factors, we focused on Egr1, encoded by an immediate–early gene and previously implicated in the proestrus-associated transcriptional regulation in the prefrontal cortex[17]. Consistent with its role as an oestrogen-responsive, immediate–early gene in the ventral hippocampus, we found higher *Egr1* expression in proestrus than in dioestrus (Fig. 3b; $P < 0.001$; Tukey's post hoc test). The pathway analysis of the proestrus-specific genes containing the Egr1 binding site (Supplementary Data 1) highlighted an overrepresentation of genes belonging to the MAPK signalling, focal adhesion, calcium signalling and the neurotrophin signalling pathways (Fig. 3c; Supplementary Data 5), consistent with an important role of Egr1 in neuroplasticity, and suggesting possible mechanistic insights into oestradiol-induced dendritic spine changes[5]. Among proestrus-specific, Egr1 motif-containing genes, we selected *Ncan*[35] (encoding Neurocan, an extracellular matrix glycoprotein), *Gria3*[36] (encoding Glutamate ionotropic receptor

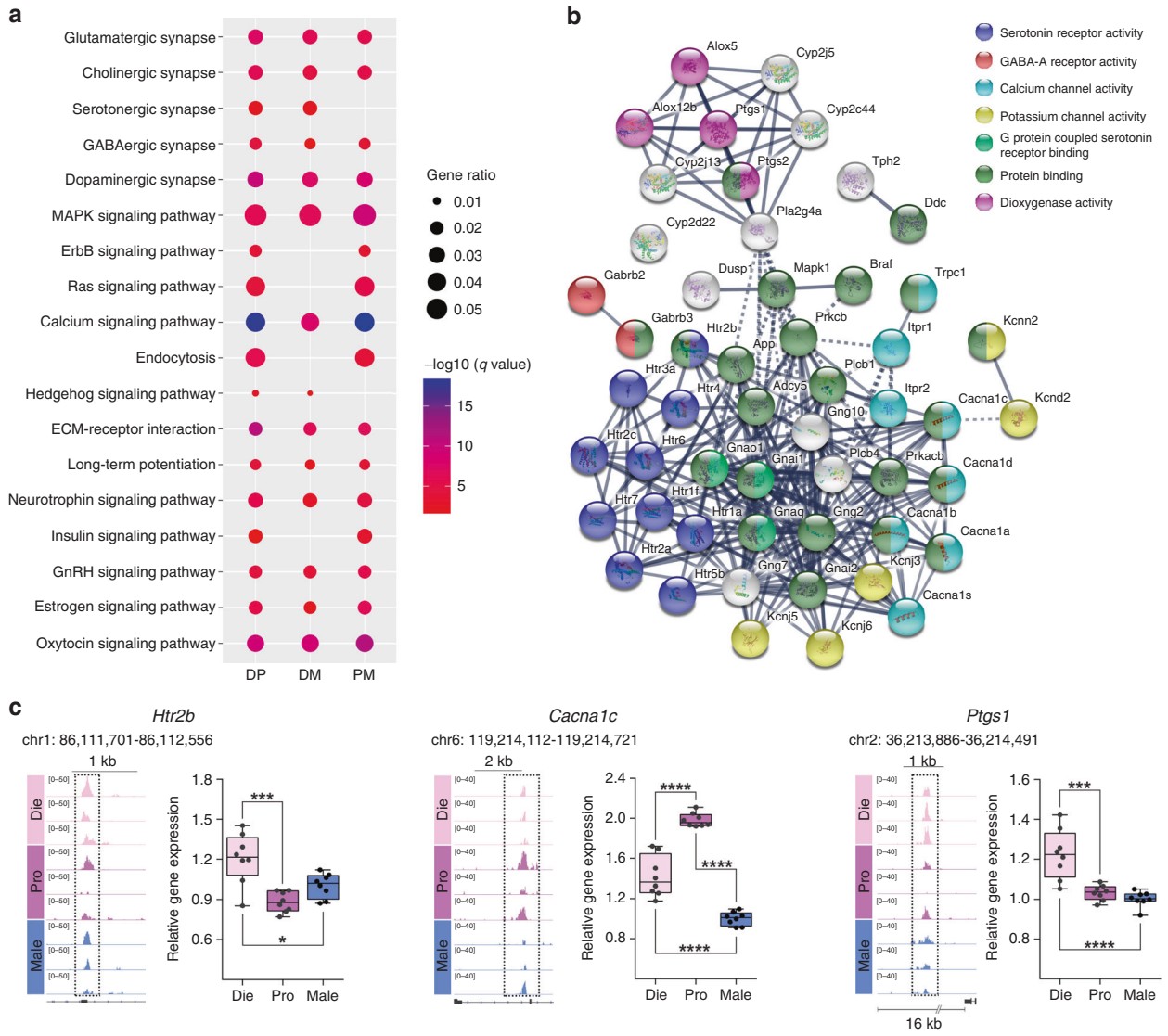

**Fig. 2** Chromatin accessibility and expression of serotonergic genes are oestrous cycle- and sex-dependent. **a** KEGG pathway analysis of the genes showing differential chromatin accessibility between the groups: DP, dioestrus–proestrus comparison; DM, dioestrus-male comparison; PM, proestrus-male comparison (note: for this analysis, all genes showing differential chromatin accessibility between the two groups were merged, e.g DP = DP_Die+DP_Pro; colours indicate adjusted p-values and dot size corresponds to gene ratio; only selected pathways are shown; n = 3 biological replicates/group). **b** Represented are all genes from the mouse KEGG serotonergic synapse pathway (mmu04726) showing differential chromatin accessibility in both DP and DM comparisons (connections between gene products are shown using String with the colour-code based on the GO molecular function). **c** Chromatin accessibility profiles of the putative regulatory regions of Htr2b, Cacna1c and Ptgs1 (n = 3 biological replicates/group) and the corresponding mRNA levels (n = 8 animals/group). Shown are the genomic coordinates of the differential ATAC-seq peaks and their relative distance to the TSS. Box plots (box, 1st–3rd quartile; horizontal line, median; whiskers, min/max); *p < 0.05; ***p < 0.001; ****p < 0.0001 (one-way ANOVA with the Tukey's post hoc test). Die (light pink), dioestrus; Pro (purple), proestrus; Male (blue), males

AMPA subunit 3), and Ptprt[37] (encoding Protein tyrosine phosphatase, receptor type T), since these genes have been implicated in dendritic spine dynamics and long-term potentiation, as well as in mood disorders. We also found that the expression of these three genes is different among the three groups (Fig. 3d; Ncan, $F_{(2,21)} = 61.38$, $P < 0.001$; Gria3, $F_{(2,21)} = 48.33$, $P < 0.001$; Ptprt, $F_{(2,21)} = 483.17$, $P < 0.001$; one-way ANOVA). Importantly, the Egr1 motif-containing peaks upstream or close to the TSS of Ncan, Gria3 and Ptprt were found in proestrus but not in dioestrus and males, and this chromatin accessibility was associated with higher mRNA levels of all three genes in proestrus compared to the other two groups (Fig. 3d).

Finally, we showed that dendritic spine density and number of synaptic connections in the ventral hippocampus are increased in proestrus compared to dioestrus and males (Fig. 3e), which further links differential chromatin accessibility, gene expression variation and changes in synaptic plasticity. Our data, which are consistent with previous studies in the dorsal hippocampus[5,38], therefore, reveal dendritic spine changes across the oestrous cycle in the hippocampal subregion associated with emotion regulation.

**Nuclear RNA expression varies with the oestrus cycle.** To further explore the functional significance of the oestrus cycle- and sex-dependent chromatin organizational changes, we performed

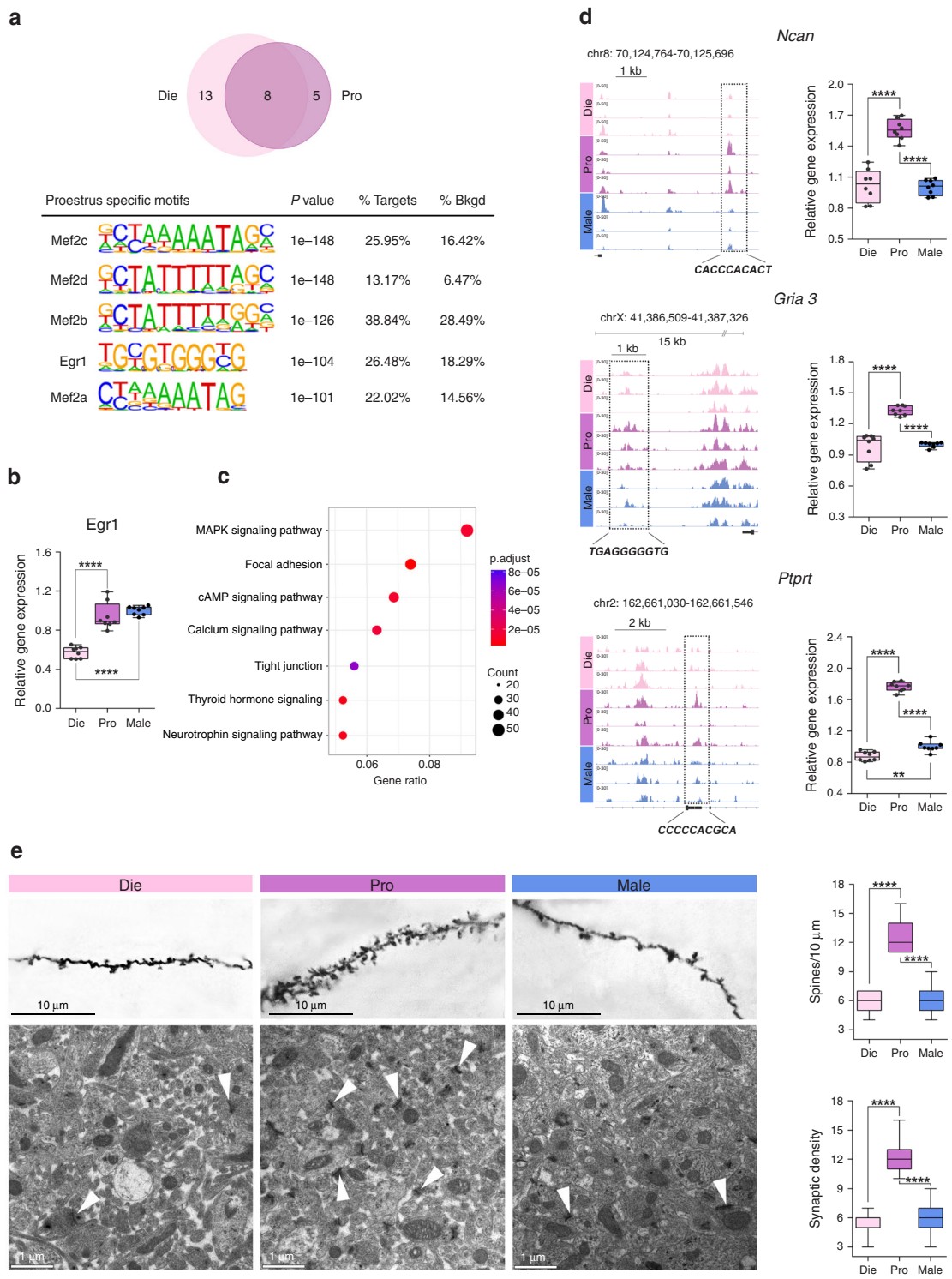

**Fig. 3** Egr1 motif-containing genes show chromatin reorganization and are implicated in the oestrous cycle- and sex-dependent synaptic plasticity. **a** Motif analysis of genomic sequences showing differential chromatin accessibility between dioestrus and proestrus identified five binding motifs as being overrepresented among the regions open in proestrus and closed in dioestrus ($n = 3$ biological replicates/group). **b** Egr1 mRNA levels in the ventral hippocampus of proestrus, dioestrus and males ($n = 8$ animals/group). **c** Top KEGG pathways enriched in Egr1 motif-containing, proestrus-specific peaks. **d** Chromatin accessibility profiles of the putative regulatory regions of Ncan, Gria3 and Ptprt ($n = 3$ biological replicates/group) and the corresponding mRNA levels ($n = 8$ animals/group). Shown are the genomic coordinates of the differential ATAC-seq peaks, their relative distance to the TSS, and a predicted Egr1 binding sequence within each peak. **e** Dendritic spine (Golgi staining, $n = 5$ animals/group; scale bar: 10 μm) and synaptic (electron microscopy, $n = 3$ animals/group, scale bar: 1 μm) density in the ventral hippocampus of dioestrus females, proestrus females, and males, and their quantification. Box plots (box, 1st–3rd quartile; horizontal line, median; whiskers, min/max); **$p < 0.01$; ****$p < 0.0001$ (one-way ANOVA with the Tukey's post hoc test). Die (light pink), dioestrus; Pro (purple), proestrus; Male (blue), males

the nuclear RNA sequencing (nucRNA-seq) analysis of purified neuronal nuclei isolated from the ventral hippocampus from proestrus, dioestrus and male animals. For many neuroscience studies, including ours, the nucRNA-seq analysis offers several advantages over a more traditional, whole-cell RNA-seq profiling typically performed on bulk brain tissue. The nucRNA-seq method (i) allows a neuronal cell-specific RNA analysis from frozen brain samples lacking intact cell membranes[39]; (ii) avoids underrepresentation of certain neuronal subtypes[40] and the spurious gene activation[39] inherent to methods requiring cellular disassociation; and (iii) has been proposed to provide a better transcriptional output to correlate with epigenomics data[39–41]. However, there are also some limitations to this method. Although the majority of expressed genes are detected in both cells and nuclei, typically the detection rate is higher in individual whole cells than in nuclei[40]. In addition, nuclear- and whole-cell-based RNA-seq methods have been shown to identify distinct groups of differentially expressed genes between the brain samples compared, yielding complementary insights into gene expression differences[42]. We therefore performed nucRNA-seq to generate a complementary data set for our ATAC-seq data on sorted neuronal nuclei, and for our whole-cell, candidate gene expression analyses on bulk ventral hippocampal tissue (Figs. 1–3).

To identify patterns of differential nuclear RNA expression among the three groups, we first generated lists of all differentially expressed genes in all comparisons (Supplementary Data 6), which included 172 genes in the proestrus–dioestrus comparison; 1367 genes in the dioestrus-male comparison and 1828 genes in the proestrus-male comparison, followed by GO and KEGG pathway analyses (Supplementary Data 7 and 8). In the proestrus–dioestrus comparison, we found enrichment of genes important for neuronal excitability, neurotransmission and synapse formation, further indicating that neuronal function across the oestrous cycle is broadly controlled by transcriptional regulation (Fig. 4a). Not surprisingly, we also found that the expression of genes involved in hormone-mediated signalling and response to hormones fluctuates across the oestrous cycle (Fig. 4a). However, less expected was the finding that the expression of chromatin regulators is cyclical in females (Fig. 4a). Several chromatin-related GO terms including nuclear chromatin, histone deacetylase complex, SWI/SNF superfamily-type complex and NuRD complex were found to be enriched in the proestrus–dioestrus comparison (Fig. 4a), and were importantly specific for the within-female comparison (Supplementary Data 7). Indeed, although gene lists for the dioestrus-male and proestrus-male comparisons are much longer and produced many more significantly enriched GO terms, including those related to specific neurotransmitter systems, subcellular compartments and behaviours, no single term in these comparisons was related to chromatin regulation and remodelling (Supplementary Data 7). We also note that genes related to fear response were enriched in the proestrus–dioestrus comparison (Fig. 4a), in line with our behavioural data.

The KEGG pathway analysis revealed several important pathways to be significantly enriched in the proestrus versus dioestrus comparison, including cAMP, Wnt, Thyroid hormone, Hedgehog and Notch signalling pathways (Fig. 4b). Importantly, in terms of gene expression, the Hedgehog signalling pathway is specific to the proestrus–dioestrus comparison (Supplementary Data 8), and considering a similar enrichment found in the corresponding ATAC-seq data (Fig. 2a), our findings suggest that the oestrous cycle-dependent regulation of genes belonging to the Hedgehog pathway involves the control of local chromatin accessibility.

To explore candidate genes of relevance to anxiety behaviour and chromatin organization, we selected some of the top genes from the

proestrus–dioestrus comparison: *Lamp5* (encoding Lysosome-associated membrane protein 5), *Dkkl1* (encoding Dickkopf-like acrosomal protein 1), *Chd3* (encoding Chromodomain helicase DNA binding protein 3) and *Smarcc2* (encoding SWI/SNF related, matrix associated, actin-dependent regulator of chromatin subfamily C member 2), and validated their differential nuclear RNA expression in ventral hippocampal neurons of proestrus and dioestrus females (Fig. 4c). *Lamp5* is a brain-specific LAMP family member, specifically implicated in trafficking and sorting of synaptic proteins, as well as in synaptic plasticity in GABAergic neurons[43]. We found this gene to be of particular interest because *Lamp5* mutant male mice show decreased anxiety-like behaviour compared to their wild-type counterparts[43]. In our study, we found that *Lamp5* shows lower expression in proestrus than in dioestrus females (Fig. 4c), which is consistent with lower anxiety levels in proestrus compared to dioestrus (Fig. 1b) and with the behavioural phenotype of the *Lamp5* $^{-/-}$ mice[43]. Another gene of interest was *Dkkl1*, which was previously identified as a hub gene in the ventral hippocampus, regulating the activity of a gene network implicated in stress susceptibility[44]. Of note was that the overexpression of *Dkkl1* in the ventral hippocampus, our area of interest, has been shown to induce increased depression-like behaviour in response to social defeat stress in male mice[44]. In our study, we show that this gene is upregulated in dioestrus compared to proestrus females (Fig. 4c), suggesting a possible role of cycling *Dkkl1* expression in regulating anxiety levels across the oestrus cycle.

Both *Chd3* and *Smarcc2* encode ATP-dependent chromatin remodelling factors, which regulate chromatin compaction and the accessibility of nucleosomal DNA through a catalytic process powered by ATP hydrolysis[45]. Chd3 is an integral subunit of the Mi-2/NuRD chromatin-remodelling histone deacetylase complex. Interestingly, a recent study has compared Chd3 protein levels in the ventral hippocampus of male mice selectively bred for either high-anxiety behaviour or normal anxiety-related behaviour, showing that lower Chd3 levels are associated with high-anxiety phenotype[46]. In line with this, we found that *Chd3* is upregulated in proestrus compared to dioestrus females (Fig. 4c), implying that changing Chd3 levels may contribute to both chromatin reorganization and variation in anxiety behaviour during the oestrous cycle. Finally, we found that *Smarcc2*, encoding the BAF170 subunit of the SWI/SNF chromatin remodeling complex[45], is more highly expressed in proestrus than in dioestrus females (Fig. 4c). This finding further confirms that the oestrus cycle is characterised by both the fluctuation of chromatin organization and by cycling RNA expression of chromatin remodelling factors in the nucleus of ventral hippocampal neurons.

**Chromatin changes overlap with transcriptional changes.** We next went on to explore the overlap between our nucRNA-seq and ATAC-seq data for all three group comparisons (Fig. 5a; Supplementary Data 9). We found that, in each group comparison, approximately half of the genes that were differentially expressed also had at least one differential ATAC-seq peak within or upstream of the gene: proestrus versus dioestrus (50.6%, or 87 out of 172 genes, Fig. 5a); dioestrus versus males (49.5%, or 676 out of 1367 genes); and proestrus versus males (52.4%, or 958 out of 1828 genes) (Supplementary Data 9). We focused on the proestrus–dioestrus comparison, where the overlap was found in genes important for chromatin remodelling (the main gene cluster, Fig. 5a), hormone signalling and neuronal excitability, among others, and included our candidate genes *Chd3*, *Smarcc2* and *Lamp5* (Fig. 5a; Supplementary Data 9). For instance, a proestrus-specific ATAC-seq peak located 300 bp downstream of the *Smarcc2* TSS was found in the proestrus–dioestrus

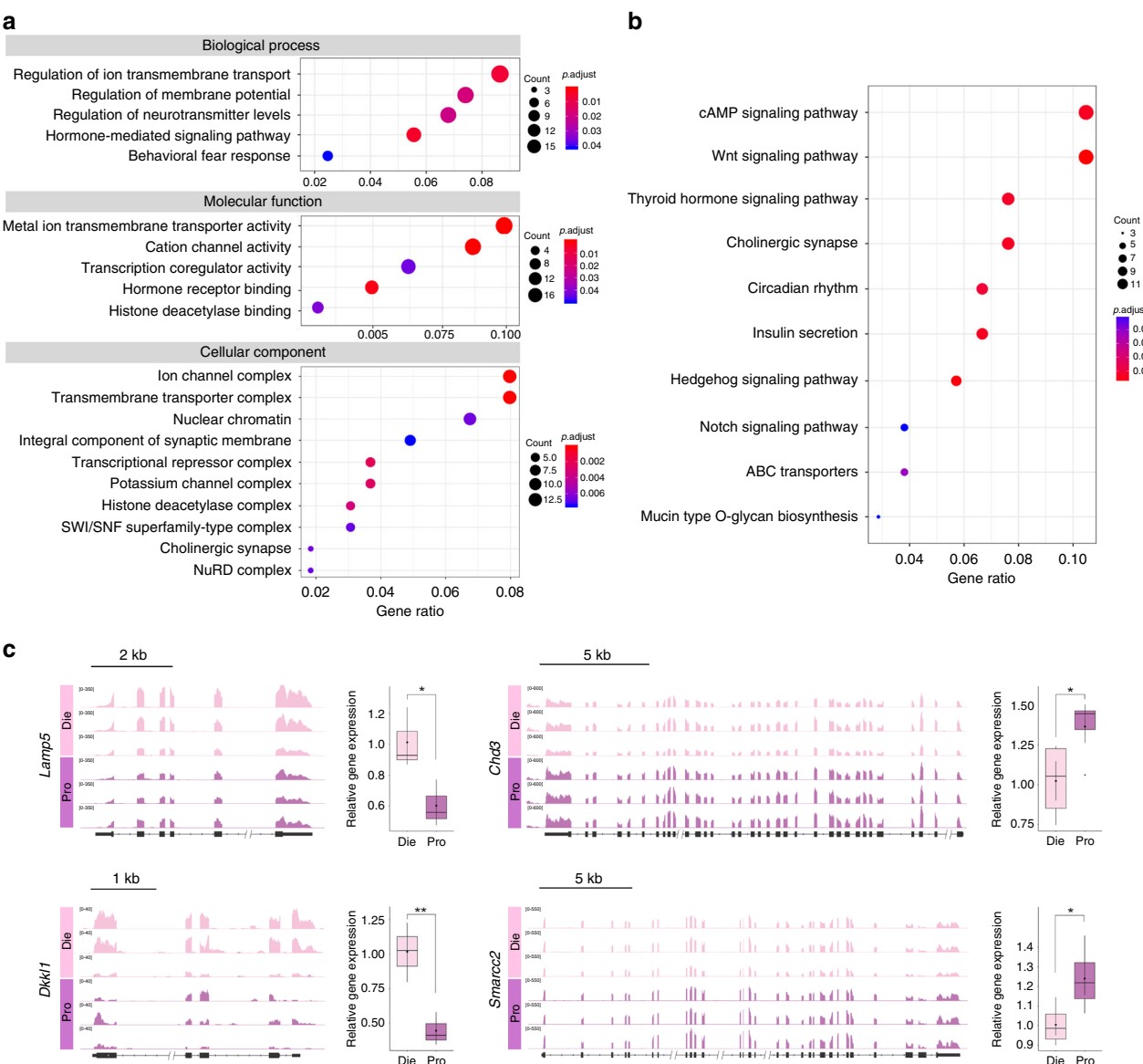

**Fig. 4** Nuclear RNA expression in ventral hippocampal neurons varies with the oestrous cycle. **a** GO analysis of genes showing differential nucRNA expression between dioestrus and proestrus (selected GO terms for biological process, molecular function, and cellular component are shown); the horizontal axis represents gene ratio, colours indicate adjusted *p*-values and dot size corresponds to gene count. **b** KEGG analysis of genes showing differential nucRNA expression between dioestrus and proestrus (ten selected pathways are shown); **c** nucRNA expression profiles of candidate genes *Lamp5*, *Dkkl1*, *Chd3* and *Smarcc2* showing differential expression between dioestrus and proestrus (shown are IGV tracks of nucRNA-seq data and corresponding box plots showing results of qPCR validation). (*n* = 3 biological replicates/group); boxplots (box, 1st–3rd quartile; horizontal line, median; whiskers, min/max; inner circles, mean ± SEM); *p < 0.05; **p < 0.01 (unpaired *t* test, one-tailed). Die (light pink), dioestrus; Pro (purple), proestrus

comparison (Supplementary Data 1) and corresponded to the increased *Smarcc2* expression in proestrus compared to dioestrus (Fig. 4c). Similarly, we found a proestrus-specific ATAC-seq peak in the vicinity (−2 kb) of the *Chd3* TSS and six dioestrus-specific ATAC-seq peaks both upstream (−144 kb, −69 kb, −61 kb, −49 kb and −45 kb) and downstream (+109 kb) of the *Lamp5* TSS (Supplementary Data 1), which corresponded to differential expression of these genes in the proestrus versus dioestrus comparison (Fig. 4c).

Next, we explored whether the overlapping genes, showing both differential gene expression and differential chromatin accessibility between proestrus and dioestrus, share any transcription factor binding site within the range of ±1 kb from the

TSS. Remarkably, this analysis revealed that the Egr1 motif was one of the top transcription factor binding sites found in the regulatory regions of 21 (24.1%) overlapping genes, including all genes of the main chromatin remodelling cluster (*Chd3*, *Chd4*, *Chd6*, *Smarrc2* and *Ncor2*) (Fig. 5b). Since the Egr1 motif was also enriched in proestrus-specific ATAC-seq peaks within the proestrus–dioestrus comparison (Fig. 3a), we checked how many overlapping genes harbour an Egr1 binding site within the region of differential chromatin accessibility. This analysis retrieved ten genes (Supplementary Data 10), among which we selected and validated *Dlk1* (encoding Delta like non-canonical notch ligand 1) as being more highly expressed in proestrus compared to dioestrus females (Fig. 5c). Consistent with this, we found two proestrus-

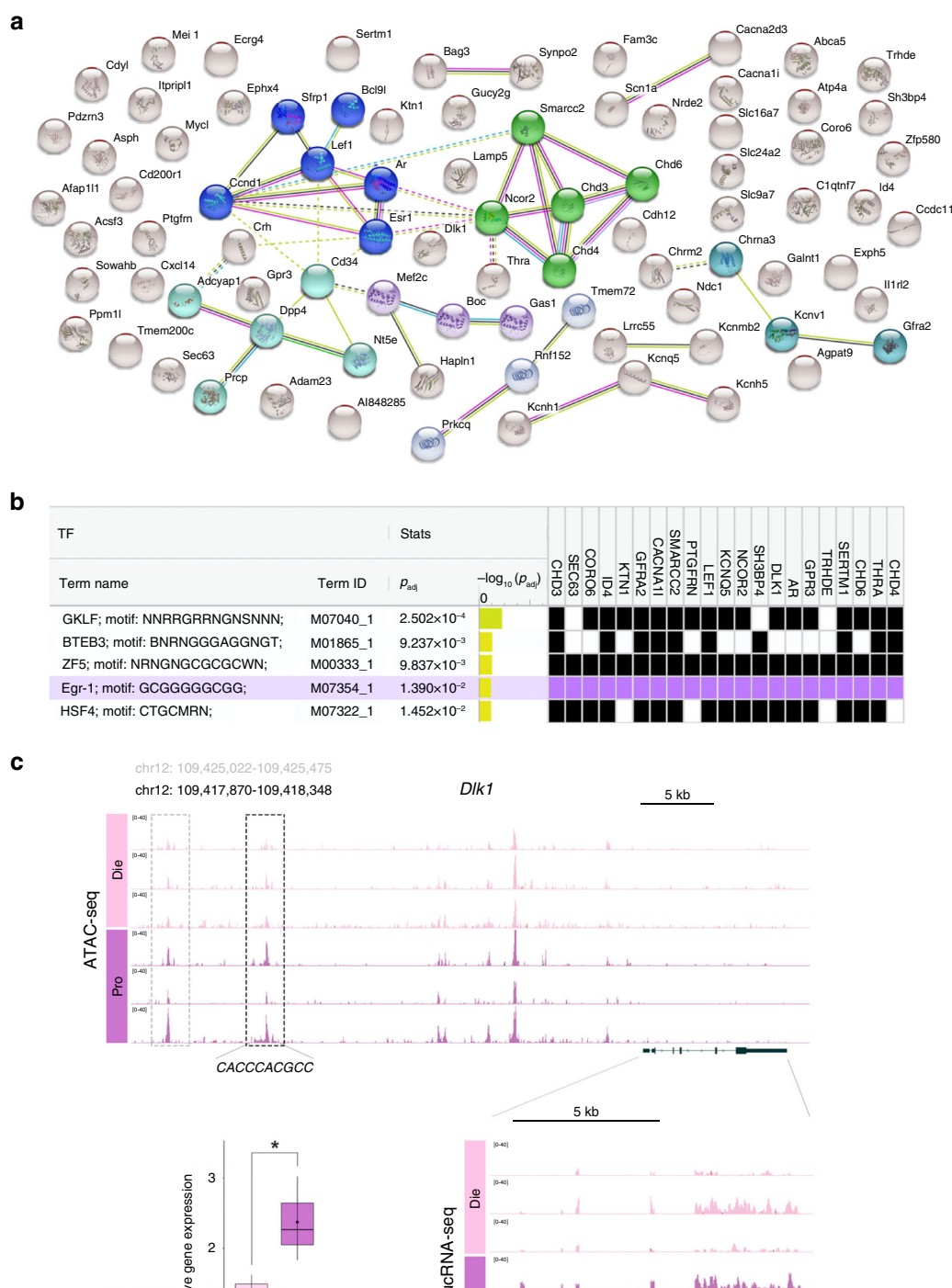

**Fig. 5** Chromatin remodelling- and anxiety-relevant genes show differential expression and varying chromatin accessibility across the oestrous cycle. **a** Shown are all genes with differential expression and local changes in ATAC-seq peak formation in the proestrus–dioestrus comparison; the connections between gene products highlighting five major gene clusters are shown using String. Of note is the main cluster corresponding to the chromatin remodelling factors (green). **b** The top five transcription binding sites identified in the vicinity of the TSS of the above genes (focused on the genes that contain an Egr1 binding site). **c** Chromatin accessibility (ATAC-seq) and nucRNA expression profile of *Dlk1* in dioestrus and proestrus. Shown are the genomic coordinates of the two differential ATAC-seq peaks, predicted Egr1 binding sequence within the second ATAC-seq peak, the IGV track of the nucRNA-seq data, and a boxplot showing qPCR validation of the nucRNA-seq data. ($n = 3$ biological replicates/group); boxplots (box, 1st–3rd quartile; horizontal line, median; whiskers, min/max; inner circles, mean ± SEM); *$p < 0.05$ (unpaired $t$ test, one-tailed). Die (light pink), dioestrus; Pro (purple), proestrus

specific ATAC-seq peaks upstream of the *Dlk1* gene, positioned at −35 kb and −28 kb from the TSS, with the latter peak also containing an Egr1 binding motif (Fig. 5c; Supplementary Data 1). This gene also harbours an Egr1 binding site in the vicinity of the TSS (Fig. 5b), which further suggests that Egr1 is involved in chromatin opening and increased transcriptional activity of *Dlk1* during the high-oestrogenic, proestrus phase.

It is also plausible that chromatin closing and downregulation of *Dlk1* expression in ventral hippocampal neurons contribute to the high-anxiety phenotype observed during dioestrus (Figs. 1b and 5c). *Dlk1* is an imprinted, paternally expressed gene, which is epigenetically regulated and strongly implicated in embryonic development[47]. Only recent studies have demonstrated the role of Dlk1 in postnatal neurogenesis[47], and we know that *Dlk1* expression in the adult brain is very restricted, largely to the reward system[47,48]. However, this gene has been recently identified as a specific transcriptional marker of the ventral hippocampus[49], with no detectable expression in other hippocampal areas, suggesting that Dlk1 may have a specific role in the regulation of emotion and reward. Consistent with this, a recent study has shown that *Dlk1* deletion in male mice is associated with increased anxiety-like behaviours in the elevated-plus maze, open-field and light–dark box tests[48]. Remarkably, dioestrus females show high anxiety in all these tests (Fig. 1b), and have significantly downregulated *Dlk1* expression compared to proestrus females (Fig. 5c). Notably, these findings indicate that neuronal expression of an imprinted gene is dynamically regulated in the adult female brain, via chromatin reorganization, likely contributing to varying anxiety levels across the oestrous cycle.

In summary, we found that genes broadly relevant to neuronal function and regulation and, in particular, relevant to chromatin regulation and anxiety behaviour, exhibit differential expression and varying chromatin accessibility across the oestrous cycle. This underscores the importance of dynamic chromatin regulation for female brain function. Considering that only a subset of genes with cycling chromatin states show differential gene expression, chromatin reorganization is likely to be involved in other important regulatory functions in the female brain, beyond the control of transcriptional initiation.

## Discussion

This study reveals remarkable chromatin dynamics in the female brain in response to naturally cycling levels of sex hormones during the oestrous cycle. We show that these chromatin dynamics have an important functional role in the brain, as we link chromatin changes to changes in neuronal gene expression, anxiety-like behaviour and synaptic plasticity in the ventral hippocampus.

The female brain has been largely understudied in basic neuroscience[1–3], with the neuroepigenetics field being no exception. One of the major reasons for excluding females from studies was that females are intrinsically more variable than males, requiring higher number of animals and complicated study designs to generate reliable data[2]. Extending previous findings of others[4,5,17,21], here we show that this variability is systematic, occurring as a function of sex hormone levels, and can be tracked all the way down to the level of DNA packaging in neuronal cells. Importantly, if we had chosen to study only sex as a variable, without taking into account the oestrous cycle stage, we would have found no statistically significant sex difference in any of the examined behavioural or structural phenotypes, with a largely distorted picture of the chromatin variability among sexes. Our approach enabled us to discover a sex hormone-induced epigenomic programming that occurs over days in post-mitotic

neurons of the female brain. As such, our findings represent an important, sex-specific addition to the recent finding in male animals showing that neuronal chromatin organization in adult brain is extensively altered in response to neuronal activity[50]. Our data further highlight the necessity to study the female brain in its naturally changing hormonal environment in order to understand how brain functions in both sexes.

Our results also provide an important insight into the mechanism that mediates oestrous-cycle-induced chromatin changes. Hormonal profiles of the oestrous cycle stages are complex, and it is likely that different hormones including oestrogen, progesterone and gonadotropins may all contribute to varying phenotypes that we observed. Here, we focused on oestrogen due to its established role in the structural[5] and behavioural[31] changes across the oestrous cycle. First, our data indicate that chromatin changes are, at least in part, induced by membrane oestrogen receptors. While genomic effects of sex hormones are traditionally expected to be mediated by classic nuclear receptors, the membrane oestrogen receptors have a prominent role in the fast oestrogen signalling in the hippocampus[51] which is likely to include indirect genomic effects observed in our study. Second, we found that an oestrogen-responsive, immediate–early gene product, Egr1, is involved in the oestrous cycle-dependent chromatin regulation and transcriptional initiation. Importantly, our results are consistent with the findings by Duclot and Kabbaj[17], which indicated that Egr1 is a major transcriptional regulator in the prefrontal cortex during the high-oestrogen, proestrus phase. Transcription factors encoded by immediate–early genes have been proposed as initiators of chromatin opening[50,52], supporting our model in which oestrogen-induced Egr1, likely via membrane-bound receptors, may be an important mediator of changes in chromatin accessibility within the proestrus phase (Supplementary Fig. 3). We also show that the expression of ATP-dependent chromatin regulators, all putative targets of Egr1, is increased in proestrus, providing an additional insight into mechanisms driving chromatin reorganization across the oestrus cycle. Additional studies are warranted to further dissect the upstream regulators and other mechanistic aspects of the oestrous cycle-dependent chromatin changes.

One of the most important implications of our findings concerns better understanding of the increased female vulnerability to anxiety and depression. Epigenetic mechanisms have been strongly implicated in these disorders[53], but the majority of mechanistic, animal studies included male rodents only. Although most recent studies included both sexes and showed large sex differences in molecular markers of depression in both humans and rodents[54,55], studies of anxiety and depression have rarely taken into account the contribution of the changing hormone status in females. From human studies, we know that the increased female risk for these disorders is strongly associated with hormonal fluctuations, as evidenced by the jump in risk coinciding with the onset of menarche and perimenopause and during the post-partum period. A subset of women also suffer from premenstrual dysphoric disorder and, strikingly, women with major depression, although under antidepressant treatment, often experience worsening of their symptoms in the low-oestrogenic, premenstrual phase of the cycle[56,57]. Here, we provide a mechanistic insight into how this female vulnerability may be mediated. We reproduce sex- and oestrous cycle-dependent anxiety behaviour in our mouse model, showing that a physiological drop in oestrogen in females results in higher anxiety levels, both compared to their high-oestrogenic phase and to males. We further show that the oestrous cycle stage significantly affects chromatin organization and expression of

genes relevant to serotonergic transmission and anxiety behaviour. It has been known that fluctuating oestrogen levels affect serotonergic function[28], but here we provide an epigenetic transcriptional mechanism contributing to these changes. In addition, it is particularly interesting to note that our candidate genes revealed by the nucRNA-seq analysis such as *Lamp5*, *Dkkl1*, *Chd3* and *Dlk1* are shown to confer vulnerability to anxiety- and depression-related behaviours in males under extreme conditions, such as genetic deletion or over-expression, chronic stress exposure, or selective breeding[43,44,46,48]. However, the expression of these same genes naturally cycles in females, reaching behavioural risk-associated levels at the time when endogenous oestrogen levels drop, and likely contributing to inherent, female-specific vulnerability to depression and anxiety disorders. Our study further reveals candidate genes and pathways, such as chromatin remodelling factors and the Hedgehog signalling pathway, opening directions for studying mechanisms underlying sex differences in anxiety and depression.

In conclusion, we found that chromatin undergoes significant reorganization within the female brain across the oestrous cycle, linked to changes in brain structure and behaviour. This study creates a foundation for a better understanding of female-biased disorders, such as anxiety and depression. Such an understanding is essential in the development of sex-specific treatments, addressing some of the most frequent and debilitating human disorders.

## Methods

**Animals and study design**. Four-week-old male and female C57BL/6J mice were obtained from the Jackson Laboratory and housed in same-sex cages ($n = 5$ per cage). Animals were maintained on a 12:12 h light:dark cycle (lights on at 8 a.m.) with ad libitum access to food and water. After 2 weeks of habituation, the oestrous cycle stage in females was assessed daily (9.00–11.00 a.m.) for the duration of three cycles, from 6 to 8 weeks of age. Each mouse cycle typically lasts 4–5 days, including proestrus, oestrus, metoestrus and dioestrus (Supplementary Fig. 1). Determination of the cycling pattern of each female (i) ensured that only females with regular cycles were included in the study; (ii) allowed precise predictions of the oestrous cycle stage for the following tests. Once three cycles were completed (at 8 weeks of age), females were placed in either proestrus or dioestrus group and tested with age-matched males. We selected proestrus and dioestrus phases as they mimic the human follicular and luteal phase of the menstrual cycle, respectively (Fig. 1a).

Behavioural tests were performed from 8 to 10 weeks of age and separated by one oestrous cycle, so that each female was always tested in the same oestrous cycle stage (proestrus or dioestrus) based on its assigned group. We confirmed the oestrous cycle stage after each behavioural test and, if the stage was not as predicted, the animal was removed from the study (this occurred in <5% of cases). After one additional cycle (at ~11 weeks of age), behaviourally tested female mice were killed at their assigned oestrous cycle stage and the stage was confirmed post mortem. Brain tissue of these animals together with the tissue of aged-matched males was used for whole-cell gene expression analysis. The ATAC-seq and nucRNA-seq analyses were performed on brain tissue isolated from 11-week-old female and male littermates that were not behaviourally tested. The oestrous cycle stage of these female animals was confirmed at each time their littermates were behaviourally tested, as well as after the animals were killed (Supplementary Fig. 1a). Histological and hormone analyses were performed on a separate cohort of animals killed at 11 weeks of age to parallel molecular analyses. All animals were killed by cervical dislocation, the brain was removed following decapitation and the ventral or whole hippocampi were immediately dissected. The hippocampal samples were either snap-frozen in liquid nitrogen and stored until use (for molecular and hormone analyses) or immediately fixed (for histological analyses). For hormone analysis, trunk blood was collected at the same time as the brain, and sera were stored at −80 °C until assayed. All animal procedures were performed in accordance with National Guidelines on the Care and Use of Laboratory Animals and a study protocol approved by the Institutional Animal Care and Use Committee at Fordham University.

**Oestrous cycle determination**. The oestrous cycle stage was assessed by the cytological analysis of vaginal smears[58]. The rounded tip of a disposable pipette was filled with 100 μl of sterile distilled water; the tip was gently placed at the opening of the vaginal canal and vaginal smear cells were collected by lavage. Vaginal smears were placed on a dry microscopic glass slide, stained with 0.1%

crystal violet staining solution for 1 min, washed and examined under the light microscope. The oestrous cycle stage (proestrus, oestrus, metoestrus and dioestrus) was monitored based on the relative ratio of nucleated epithelial cells, cornified squamous epithelial cells and leukocytes present in vaginal smears (Supplementary Fig. 1b).

**Hormone analysis**. To confirm that vaginal cytology closely reflects sex hormone levels in serum and the hippocampus (proestrus: high oestradiol–low progesterone; dioestrus: low oestradiol–high progesterone), we collected vaginal smears (to determine the oestrous cycle stage) and serum and the hippocampus (to measure sex hormone levels) from the same animals post mortem (Supplementary Fig. 1c). Serum and hippocampal levels of oestradiol and progesterone were determined by commercial mouse ELISA kits, using materials and protocols supplied by BioVision (K3830) and BioVendor (RTC008R), respectively. For the serum hormone analysis, trunk blood was collected into 1.5-ml tubes, allowed to coagulate spontaneously at room temperature for a 2-h period, and serum was then separated by centrifugation ($1503 \times g$, 20 min, 4 °C) and stored at −80 °C until assayed. For the hippocampal hormone analysis, snap-frozen hippocampal tissue was homogenised in 0.05 M Tris-sucrose buffer (pH 7.9) on ice using a manual tissue homogeniser. The homogenate was centrifuged at high speed ($106,882.9 \times g$, 1 h, 4 °C, Beckman ultracentrifuge), supernatant was collected and stored at −80 °C until assayed. All serum and hippocampal samples (50 μl of 5 ×-diluted sample for oestradiol; 25 μl of undiluted sample for progesterone) were measured together with oestradiol or progesterone standards in duplicate with an intra-assay coefficient of variation <10%. The absorbance was measured using a plate reader with a 450-nm filter (SpectraMax, Molecular Devices).

**Behavioural tests**. All behavioural testing occurred during the animals' light cycle. Females in the proestrus group ($n = 14$–16) were only tested during their proestrus stage (9:00–11:00 a.m., when oestradiol levels are high and progesterone levels are low), while females in the dioestrus group ($n = 14$) were only tested in the morning of the first day of dioestrus (9:00 a.m.–12 p.m., when oestradiol is low and progesterone is high). Males ($n = 12$) were age-matched and tested in parallel (10:00 a.m.–1:00 p.m). Tests for anxiety-like behaviour were performed in the following order: open-field; light–dark box; and elevated-plus maze test. For all tests, behaviours were video-recorded; animals were tracked and data were analysed by the ANY-maze Video Tracking Software (ANY-maze 5.1, Stoelting Co, IL).

For the open-field test, the chamber consists of a square acrylic box ($40 \times 40 \times 35$ cm) with clear walls and grey base plate (Stoelting). On the day of testing, each mouse was placed in the lower left corner of the apparatus and allowed to freely explore the open field for 10 min. For tracking purposes, the open field was divided into the centre zone ($20 \times 20$ cm) and the periphery. Behaviours scored included: time spent in the centre; number of entries into the centre zone; time spent in the periphery; and total distance travelled.

For the light–dark box test, the apparatus ($40 \times 40 \times 35$ cm) is partitioned into a brightly lit chamber ($20 \times 40$ cm) and a closed, black chamber ($20 \times 40$ cm) connected by a $7.5 \times 7.5$ cm door in the middle of the wall separating the two chambers (Stoelting). Each mouse was placed at the door and allowed to move freely between the two chambers. During a 10-min session, the number of transitions between the light and dark compartments and time spent in each compartment were recorded.

For the elevated-plus maze test, the apparatus (Stoelting) consists of a grey-coloured, plus-shaped maze with two open arms ($35 \times 5$ cm) and equally sized closed arms with grey walls (15-cm high). The arms are connected by a centre platform ($5 \times 5$ cm) and the maze is elevated 50 cm off the ground. On the day of testing, each mouse was placed in the central platform facing an open arm and allowed to freely explore the maze. During a 5-min session, the following behaviours were scored: total time spent in the open arm, total time spent in the closed arm and total distance travelled.

**Chromatin accessibility assay of purified neuronal nuclei**. Total nuclei isolation and purification of neuronal nuclei were performed using our previously established protocol[22]. Briefly, six animals per each group—proestrus, dioestrus and males (total $n = 18$)—were used for chromatin accessibility analysis; the ventral hippocampus was randomly dissected from one side of the brain from each animal (equally representing the left and right hippocampi), then frozen in liquid nitrogen and later processed. Nuclei preparation and sorting were performed in three batches ($n = 6$) with each batch having equal group distribution ($n = 2$ for proestrus, dioestrus and males). Each ventral hippocampus was cut into small pieces, and homogenised by douncing. Total nuclei were extracted using sucrose-gradient ultracentrifugation ($101,814.1 \times g$, 1 h, 4 °C, Beckman ultracentrifuge). The nuclei were resuspended in DBPS and immunolabeled with the mouse monoclonal antibody against neuronal marker NeuN conjugated to AlexaFluor 488 (1:1000; Millipore, MAB377X). Prior to sorting, DAPI (1:1000; Thermo Fisher Scientific, 62248) was added to the incubation mixture and samples were filtered through a 35-μm cell strainer. Fluorescence-activated nuclei sorting (FANS) was performed on a FACSAria instrument (BD

Sciences) at the Albert Einstein College of Medicine Flow Cytometry Core Facility. To set up the experimental protocol, we used three controls: (i) DAPI only; (ii) IgG1 isotype control-AlexaFluor 488 and DAPI; and (iii) NeuN-AlexaFluor 488 only; in addition to a sample containing NeuN-AlexaFluor 488 and DAPI stain (Supplementary Fig. 5a). For each sample, gates were adjusted to (i) select nuclei from debris; (ii) ensure single nuclear sorting (using DAPI); and (iii) select the NeuN+ (neuronal) and NeuN− (non-neuronal) nuclei populations (Supplementary Fig. 5b). From each individual ventral hippo-campus, we collected 30,000 NeuN+ (neuronal) nuclei in BSA-precoated tubes filled with 200 µL of DPBS. The purity of sorted nuclei was confirmed using fluorescent microscopy (Supplementary Fig. 5c).

Chromatin accessibility (ATAC-seq) assay was performed according to Buenrostro et al.[24], with some modifications. Following FANS, 30,000 neuronal nuclei from the ventral hippocampus were spun down (2,900 × g, 10 min, 4 °C), the supernatant was removed and nuclei pellet was resuspended in 50 µL of the transposase reaction mix including 25 µL 2xTD reaction buffer and 2.5 µL of Tn5 Transposase (Nextera DNA Library Preparation Kit, Illumina, FC-121-1030). The transposition reaction was performed at 37 °C for 30 min, followed by purification using MinElute PCR Purification Kit (Qiagen, 28004). A purified, transposed DNA was eluted in 10 µL of EB elution buffer and stored at −20 °C until amplification. The library preparation was performed in two batches of nine samples with equal group distribution (n = 3 for proestrus, dioestrus and males). For indexing and amplification of transposed DNA, we combined the following for each sample: 10 µL of transposed DNA, 5 µL of PCR Primer Cocktail (PPC, Illumina FC-121–1030), 25 µl of NEBNext High-Fidelity 2x PCR Master Mix (New England Biolabs, M0541S) and 5 µL each of Nextera i5 and i7 indexed amplification primers (Nextera Index Kit, Illumina, FC-121–1011). The PCR reaction was carried out using the following conditions: one cycle of 72 °C for 5 min and 98 °C for 30 s; followed by five cycles of 98 °C for 10 s, 63 °C for 30 s and 72 °C for 1 min; and a hold step at 4 °C. We then performed a 20-cycle qPCR side reaction to determine the optimal number of PCR cycles, combining 5 µL of a previously PCR-amplified DNA with 0.5 µL PPC, 5 µl of NEBNext High-Fidelity 2x PCR Master Mix, and 4.5 µL of 0.6 × 2xSYBR green I (Invitrogen, S7563). For our experiments, four to five additional PCR cycles were added to the initial set of five cycles to ensure optimal amplification and obtain high-complexity libraries. The libraries were purified using MinElute PCR Purification Kit and eluted in 20 µL of the elution buffer. The library quality was assessed by 2.2% agarose gel electrophoresis (FlashGel DNA System, Lonza) and Bioanalyzer High-Sensitivity DNA Assay (Supplementary Fig. 6). The ATAC-seq libraries were quantified by Qubit HS DNA kit (Life Technologies, Q32851) and the qPCR method (KAPA Biosystems, KK4873) prior to sequencing. 100 bp, paired-end sequencing was performed on a HiSeq 2500 Illumina instrument at the Albert Einstein College of Medicine Epigenomics Shared Facility. Total of 18 ATAC-seq libraries were sequenced in three lanes with equal group distribution (six libraries/lane; n = 2 for proestrus, dioestrus and males).

**ATAC-seq data analysis**. To reduce technical variability, including batch effects and read number variation, we first combined the obtained sequences by sorting batch (n = 3 per group). Next, the sequences were adapter trimmed and aligned to mouse reference genome (mm10 including small contigs) using BWA-MEM software[59]. For the analysis, only reads uniquely aligned to mm10 canonical chromosomes were used, excluding mitochondrial DNA. The peak-calling was performed using MACS2 as previously reported[24]. We observed a positive correlation between read numbers and detected peak numbers. To correct for the peak number bias caused by unequal number of reads between samples, we performed downsampling and re-performed peak-calling on comparable number of reads for each sample. The downsampling factor was calculated based on the read numbers and RiP score (reads in peaks) from ChIPQC Bioconductor package[60]. For each sample, the original read number, quality control metrics, down-sampling factor and final peak number are shown in Supplementary Table 1. To select high-confidence peaks for each group, we calculated the irreproducible discovery rate (IDR) of the detected peaks between pairs of biological replicates[61]. We performed IDR analysis of all combinations in each group, then we selected high-confidence peaks which are shared across at least two biological replicates with IDR < 0.05. We used the high-confidence peaks for the further analysis. The annotation of peaks was performed with *annotate-PeakInBatch* function on UCSC.mm10.knownGene annotation[62], and peak overlap status was analysed with *findOverlapsOfPeaks* function of ChIPpeakAnno Bioconductor package[63]. Motif analysis was performed on the identified peaks using HOMER motif analysis (v3.12) with *findMotifsGenome.pl* function[64]. Gene Ontology and KEGG Pathway analyses were performed on the peaks located between 10 percentile (−135,672 bp) to 90 percentile (203,196 bp) of the distance from the transcription start sites of the UCSC mm10 known genes with clusterProfiler Bioconductor package[65] at significance level q < 0.05.

**Candidate gene expression analysis**. For whole-cell candidate gene expression analysis in bulk ventral hippocampal tissue (Figs. 1–3), the total RNA was isolated from the ventral hippocampi using Allprep DNA/RNA mini kit (Qiagen). Gene expression was assessed using reverse transcription (SuperScript III First-Strand Synthesis System; Invitrogen) followed by quantitative PCR with a Quant Studio 3

real-time PCR system (Applied Biosystems). Relative mRNA levels were calculated using the standard 2-ΔΔCT method with males as a reference sample and cyclo-philin A (*Ppia*) as an endogenous reference gene. Primer sequences used to quantify mRNA levels are as follows: *Ppp1r1b* (forward: 5′- GGACGAAGAAGA AGACAGCCA-3′, reverse: 5′-CACTTGGTCCTCAGAGTTTCC-3′); *Kcnv1* (for-ward: 5′-TGAACTCAGCATTGACTCCTGC-3′, reverse: 5′-ACTTTCCTGGTCA TCTGTGTCC-3′; *Syn1* (forward: 5′-CCACTGCTGAGCCCTTCATTG-3′, reverse: 5′-CAGTTACCCGACACTGATGTCC-3′); *Htr2b* (forward: 5′- AGACAGACTC AGTAGCAGAGGA-3′, reverse: 5′-GCAACAGCCAGAATCACAAGG-3′); *Cac-na1c* (forward: 5′-GGCTCTGCTGTGTCTGACC-3′; reverse: 5′-TACTCCACT CGTTCCAGGTTGG-3′); *Ptgs1* (forward: 5′-TGTTACTATCCGTGCCAGAACC-3′, reverse: 5′-GTCAGCAGGAAATGGGTGAAC-3′); *Egr1* (forward: 5′- AGCGA ACAACCCTATGAGCAC-3′, reverse: 5′-GGATAACTCGTCTCCACCATCG-3′); *Ncan* (forward: 5′- TGCGATAC CAGTGTGATGAAGG-3′, reverse: 5′-TTGTG ATGTCGGTGTGGATGG-3′); *Gria3* (forward: 5′-GGTCATTCTCACGGAGGAT TCC-3′; reverse: 5′-GAGTGGTTCTGGTTGGTGTTG-3′); *Ptprt* (forward: 5′- G TATAGCGTGGCTCTGGGAAC-3′, reverse: 5′-TGGCAGAAGAAGATGGGCTT TC-3′); *Ppia* (forward: 5′-GAGCTGTTTGCAGACAAAGTTC-3′, reverse: 5′-CC CTGGCACATGAATCCTGG-3′). For validation of candidate genes revealed by the nucRNA-seq analysis (Figs. 4–5), we used technical replicates of the same samples analysed by nucRNA-seq. cDNA from nuclear RNA samples isolated from sorted neurons was quantified using qPCR with the following primer sets: *Lamp5* (forward: 5′-GGCATGTCTTGGGGTTCG-3′, reverse: 5′-ACTTACTCCGCAGTA TGGACAC-3′); *Dkkl1* (forward: 5′- TTCGTGTCCTCCTCTGCTC-3′, reverse: 5′-AGGTGTTCTGCTGAGAGTCG-3′); *Chd3* (forward: 5′-AGGTCCCTTCCTGG TGAG-3′, reverse: 5′-TCCTTGTCGCCCGTATATGTG-3′); *Smarcc2* (forward: 5′-CGACACATTCAACGAGTGGATG-3′, reverse: 5′-ATCTTCTTCCTGCGGGA-GAC-3′); *Dlk1* (forward: 5′-CAACAATGGAACTTGCGTGGAC-3′, reverse: 5′-C CAGAGAACCCAGGTGTGC-3′); and *Ppia* (forward: 5′-AGGTCCTGGCATCTT GTCC-3′, reverse: 5′-GGGAACCGTTTGTGTTTGGTC-3′). *Ppia* was used as a reference gene for both whole-cell and nuclear RNA expression analysis, as we have confirmed that, in either case, the expression of this gene does not vary with the oestrous cycle stage or sex.

**Nuclear RNA sequencing (nucRNA-seq)**. For nucRNA-seq, nuclei isolation and purification of neuronal nuclei were performed as described for ATAC-seq. Six animals per each group—proestrus, dioestrus and males (total n = 18)—were used for the analysis. As this assay requires a larger amount of starting material, the ventral hippocampus was dissected bilaterally from each animal, and brain tissue from two animals was pooled for each biological replicate (n = 3 repli-cates/group). Nuclei preparation and sorting were performed in three batches (n = 3) with each batch having equal group distribution (n = 1 for proestrus, dioestrus and males). Per each biological replicate, 250,000 neuronal nuclei were directly sorted into Trizol LS reagent (Thermo Fisher Scientific, 10296-010) to protect nucRNA from degradation. After the addition of chloroform, the aqu-eous phase was recovered and the total nucRNA was further isolated and purified using the miRNeasy Micro kit reagents and the RNeasy MinElute spin column (Qiagen). The quality of nucRNA was assessed using Bioanalyzer RNA 6000 Pico assay (Agilent), with each sample having RIN >7 (Supplementary Fig. 7). The quantity of nucRNA was determined using Qubit RNA High-Sensitivity assay (Thermo Fisher Scientific, Q32852). After quality control, all nucRNA (16–26 ng/sample) was used to construct cDNA sequencing library using the KAPA RNA HyperPrep Kit with RiboErase (KK8560, KAPA Bio-systems), following general manufacturer's instructions. Briefly, rRNA was first depleted from the total nucRNA by hybridisation of complementary DNA oligonucleotides followed by treatment with RNase H and DNase. We con-firmed the efficiency of rRNA depletion using this method by (i) running the Bioanalyzer RNA 6000 Pico assay (Supplementary Fig. 7) and (ii) quantifying the 28S rRNA transcript by qRT-PCR before and after RiboErase treatment using the following primers: forward, 5′- CCCATATCCGCAGCAGG TC-3′; reverse, 5′-CCAGCCCTTAGAGCCAATCC-3′. rRNA-depleted nucRNA was fragmented at 94 °C for 5 min in the presence of $Mg^{2+}$, followed by the first- and second-strand cDNA synthesis and A-tailing. Following the adapter ligation with the KAPA single-indexed adapters (KK8701), each library was amplified using 13 PCR cycles. After a bead-based cleanup, the library quality was examined using Bioanalyzer High-Sensitivity DNA Assay (Supplementary Fig. 7). The libraries were quantified by Qubit HS DNA kit prior to sequencing. 100 bp, paired-end sequencing was performed on a rapid run mode of a HiSeq 4000 instrument at the New York Genome Center. A library pool containing equal amounts of all 9 libraries (n = 3 for proestrus, dioestrus and males) was prepared and loaded on two lanes, yielding close to 40 million reads per sample (Supplementary Table 2).

**nucRNA-seq data analysis**. The obtained sequences were adapter trimmed and aligned to the mouse reference genome (mm10 including small contigs) with the GENCODE M15 gtf file using Star Aligner[66]. The nucRNA-seq data basic infor-mation is summarised in Supplementary Table 2. Before the analyses, we assessed the sequence quality by FastQC (http://www.bioinformatics.babraham.ac.uk/projects/fastqc/), and the library quality by RSeQC[67], and no library was found to be of low quality. We used DESeq2[68] to identify differentially expressed genes at

significance level $p < 0.05$. The Gene Ontology and KEGG Pathway enrichment analyses were performed with the clusterProfiler Bioconductor package[65] at significance level $q < 0.05$. The analysis of transcription factor binding sites was performed with g:Profiler[69] using the TRANSFAC database, setting the threshold at significance level $q < 0.05$.

**Immunofluorescence**. Brains were fixed in cold, 4% buffered paraformaldehyde (PFA) in 0.1 M PBS (pH 7.4) for 24 h. Samples were dehydrated in increasing concentrations of sucrose (10–30%) prepared in 0.1M PBS at 4 °C for the following 6 days and then frozen on dry ice. Next, samples were embedded in the optimal cutting temperature (OCT) compound prior to sectioning, and each block of tissue containing the ventral hippocampus was serially sectioned on rotatory cryocut (Leica CM1950, Leica Biosystems GmbH). The 20-μm sagittal sections were placed on super-frost ultra-plus glass slides and processed for immunofluorescent staining. After initial tissue rehydration, heat-induced antigen retrieval was performed in 0.01 M citrate buffer (pH 6.0) using a microwave (750 W, $3 \times 7$ min). After two washes with PBS, slides were incubated with 10% normal goat serum (Abcam, ab156046) and 0.1% Triton X-100 in PBS at room temperature for 1 h. After blocking, the sections were incubated at 4 °C for 48 h with the following primary antibodies: (i) rabbit polyclonal anti-ERβ antibody (1:100; Abcam, ab3576); (ii) mouse monoclonal anti-DARPP32 antibody (1:200; Santa Cruz Biotechnology, sc-271111); and (iii) chicken anti-MAP2 antibody (1:10000; Abcam, ab5392). Primary antibodies generated in different species were used to prevent cross-reactions between primary and secondary antibodies when triple staining was performed. Following the incubation with the primary antibody, sections were incubated in dark with appropriate secondary antibodies for 2 h at 4 °C: (i) anti-rabbit IgG AlexaFluor 647 conjugate for ERβ visualisation (1:200; Abcam, ab150079); (ii) anti-mouse IgG CruzFluor 594 conjugate for DARPP32 (1:100; Santa Cruz Biotechnology, sc-516178); and (iii) anti-chicken IgY AlexaFluor 488 conjugate for MAP2 (1:200; Abcam, ab150169). Negative controls were obtained by replacing the primary antibody with PBS. After rinsing in PBS, the sections were counterstained with DAPI (1:10,000) and mounted using Mowiol 4–88 (Sigma-Aldrich). Images were acquired on a confocal laser scanning microscope Leica TCS SP5 system (Leica Microsystems CMS GmbH) using sequential scanning mode.

**Golgi-Cox Staining**. After decapitation, brains ($n = 5$/group) were briefly rinsed with distilled water and split into half along the longitudinal fissure. One half of the brain was used for Golgi-Cox staining and another was processed for electron microscopy. The Golgi-Cox staining procedure was performed using Golgi-Cox OptimStain Kit (Hitobiotec Inc., HTKNS1125) as per the manufacturer's instructions. In brief, brain sample was transferred to a small vial with premixed Golgi-Cox impregnation solution and then stored in the dark at room temperature. The impregnation solution was changed after 24 h, and brains were impregnated for additional 2 weeks. Samples were then transferred into a tissue-protectant solution and stored in the dark at 4 °C for 72 h. Samples were frozen in hexane cooled with dry ice and then embedded in the OCT compound. Each sample was slowly cut into 80-μm-thick coronal sections within the hippocampal region using a cryostat (Leica CM1950). Sections were placed on gelatin-coated slides and stained according to the protocol. Next, the sections were dehydrated in increasing concentrations of ethanol and xylene, and cover slips were mounted with DPX (Sigma-Aldrich). Bright-field images of ventral hippocampal pyramidal neurons were taken at × 63 magnification on a Zeiss Axio Observer Microscope equipped with Axiocam MRc Zeiss camera and Axiovision 4.8 Software (Zeiss). The number of dendritic spines was analysed using ImageJ (public domain software from the National Institutes of Health; http://imagej.nih.gov/ij/)[70].

**Transmission electron microscopy**. The ventral hippocampus was dissected out from half of the brain ($n = 3$/group). Each sample was cut into small pieces which were immersed in cold fixative solution (4 °C). The mixture of 2% paraformaldehyde and 2.5% glutaraldehyde in 0.1 M cacodylate buffer was used as fixative. After 24-h fixation, samples were post fixed in 1% osmium tetroxide dissolved in the same buffer for 1 h. Samples were dehydrated through a graded series of ethanol (30–100%). The following procedure, including tissue embedding in acrylic resin, was performed at the Albert Einstein College of Medicine Analytical Imaging Facility. The embedded tissue blocks were cut with diamond knives (Diatome) on Leica UC7 ultramicrotome system (Leica Microsystems). For the ultrastructural examination, ultrathin sections (70 nm) were mounted on grids, stained with uranyl acetate followed by lead citrate solution and examined with the JEOL 1200EX transmission electron microscope (JEOL Inc.) at 80 kV. Images were taken using Gatan Orius Digital 2Kx2K CCD Camera. Forty electron micrographs per group at original magnification of × 10,000 were used for ultrastructural image analyses. Synapses were identified by the presence of presynaptic vesicles and electron-dense postsynaptic densities. The synapse density was defined by the number of synapses along a 1-μm length of the neuronal membrane.

**Statistical analysis**. Behavioural, whole-cell candidate gene expression, hormone level and histological data were analysed using one-way ANOVA with the Tukey's

post hoc test using the IBM SPSS Statistics 25.0 software. Differences in candidate gene nuclear RNA expression between proestrus and dioestrus groups were analysed using $t$ test. $P$-values $< 0.05$ were considered significant.

**Reporting summary**. Further information on research design is available in the Nature Research Reporting Summary linked to this article.

## Data availability
ATAC-seq and nucRNA-seq data are available from the NCBI Gene Expression Omnibus database under accession number GSE114036. All other relevant data supporting the key findings of this study are available within the article and its Supplementary Information files or from the corresponding author upon reasonable request. A reporting summary for this article is available as a Supplementary Information file.

## Code availability
The code files for the ATAC-seq analysis are available at: https://github.com/MasakoSuzuki/Jaric_etal_2018.

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

## Acknowledgements

This study was supported by the NARSAD Young Investigator Grant from the Brain & Behavior Research Foundation to M.K. We would further like to acknowledge the resources of the Center for Epigenomics at the Albert Einstein College of Medicine. Finally, we would like to thank Lydia Tesfa for her assistance with nuclei sorting as well as Shahina Maqbool and Benjamin Hubert for their assistance with massively parallel sequencing.

## Author contributions

I.J., M.S. and M.K. designed the study; I.J. performed oestrous cycle monitoring and predictions, behavioural and hormone analysis, ATAC-seq experiments, whole-cell gene expression assays and histology; D.R. assisted with behavioural experiments and tissue collection, and designed qRT-PCR primers; D.R. and M.K. performed nucRNA-seq experiments and gene validation; I.J. and D.R. performed statistical analysis; M.S. performed bioinformatics analysis; J.M.G. contributed computational resources; I.J., J.M.G., M.S. and M.K. interpreted the data; I.J. and M.S. constructed the figures; M.K. conceived and directed the project, and wrote the main text of the article; I.J., M.S., and M.K. wrote the Methods section; J.M.G revised the paper; all authors commented on and approved the final version of the paper.

## Additional information

**Competing interests:** The authors declare no competing interests.

