## [Peer Review File · Nature Communications]

Reviewers' comments:

Reviewer #1 (Remarks to the Author):

Jaric et al. investigated differences in chromatin accessibility in ventral hippocampal neurons during the estrus cycle in female mice. To draw the comparison to anxiety, they validated a known link between dioestrus and anxiety, then followed this up by investigating differences in chromatin accessibility during the two phases. They identified genes that are uniquely accessible during diestrus vs proestrus and in each phase compared to males. The functional relevance of chromatin changes was confirmed with evaluations of gene expression in candidate genes.

To elucidate the potential mechanism underlying altered chromatin accessibility, the authors looked for ER binding sites and instead identified enrichment of binding sites that are consistent with second messenger activity of membrane-bound ER receptors, identifying a potential mechanism to explain variation in chromatin accessibility.

Overall, this study demonstrates an important comparison of hippocampal chromatin dynamics across the estrus cycle, which is important for understanding the hormonal contribution to mood disorders. This study was very well designed, well controlled, and impressively executed. It is rare that I have so few comments, but virtually every question that arose as I read the manuscript was addressed at some point, which is commendable.

There are a few experimental strategies that would be helpful additions, but to avoid acting as an additional author on the paper, these are only suggestions that I do believe are not necessary inclusions for the publication of this manuscript.

1. Given the emphasis on shifts in ER receptor activation, it would be of interest to demonstrate that membrane-bound ER is indeed the driver of these changes. There are several methods for achieving this, which include ovariectomy with and without E2 replacement. To differentiate between effects of membrane-bound and nuclear ER, it would be helpful to include BSA-bound E2 that cannot cross the membrane, thus limiting the effects to only membrane bound receptors.

Overall, this was an excellent and well designed paper.

Reviewer #2 (Remarks to the Author):

In the manuscript, entitled "Chromatin organization in the female brain fluctuates across the oestrous cycle" from Jaric and colleagues, the authors present an intriguing set of experiments aimed at increasing our understanding of the potential dynamic nature of chromatin organization/accessibility in neurons in response to hormonal fluctuations in females. Such investigations have merit, as most related studies to date have exclusively used males. Furthermore, as the authors point out, since a discordance exists between males and females in the precipitation and symptomology of numerous psychiatric syndromes including anxiety and mood related disorders, further studies in females accounting for such alterations in the oestrous cycle are desperately needed.

To accomplish their goals, the authors present behavioral and ATAC-seq data to support the following:

- a) Dioestrus females exhibit heightened anxiety-like behaviors vs. proestrus females and males, as demonstrated through alterations in open-field, light dark box and EPM behaviors.
- b) While largely overlapping, open chromatin regions (OCRs) are significantly altered in dioestrus females vs. proestrus females and males, with such changes associating with altered expression of specific target genes (as measured using candidate-based approaches).
- c) Pathways associated with 5-HTergic synapses and Hedgehog signaling are enriched in dioestrus females vs. proestrus females and males, perhaps making sense in light of known roles for 5-HT

signaling in mood disorders and their respective treatment.

d) Motif analyses indicate putative upstream regulators (e.g., Egr1) associated with alterations in chromatin accessibility in dioestrus females.

e) Dendritic spine density/numbers are altered in response to the oestrous cycle.

Overall, the manuscript is well-written and presents an interesting, yet highly descriptive, assessment of fluctuations in chromatin organization throughout the oestrous cycle. There are numerous concerns, however, that should be addressed prior to publication in Nature Communications.

1) As it stands, given its descriptive nature, this manuscript reads a lot like a Resource article, which is not necessarily a bad thing. However, if the authors choose to keep the manuscript descriptive, then they should consider expanding their analyses to include multiple brain regions in order to examine whether hormonal fluctuations result in similar or dissimilar outcomes brain-wide. To do so, the authors may elect to choose one other brain region associated with mood disorders (e.g., mPFC, ventral striatum, etc.), along with one that is not typically associated with the behaviors that they are assessing (e.g., visual cortex).

2) While their candidate-based approach to testing whether genes displaying alterations in chromatin accessibility also associate with changes in expression is reasonable, it would be helpful to understand on a more global level whether such correlations exist genome-wide. Therefore, the authors should perform RNA-seq on sorted neurons across the oestrous cycle to determine this more accurately.

3) While the upstream motif analyses are interesting, this part of the story needs to be further developed. For example, if the authors believe that Egr1 is driving a transcriptional program associated with dioestrus, then they should perform manipulations (e.g., viral OE) of Egr1, followed by ATAC-seq (or targeted ATAC-PCR) to verify this effect directly.

4) The spine data in Fig. 3e, while interesting, seem out of place in this manuscript unless the authors wish to further link manipulations in chromatin accessibility to these changes. In other words, if they want to link these phenomena, then I would suggest performing manipulations of a specific ATAC targets or upstream regulators (perhaps Egr1) to show that this will alter the differences in spines observed throughout the oestrous cycle.

Similarly, it would be nice if the authors could link their behavioral analyses in Fig. 1 to alterations in accessibility/gene expression by manipulating targets and/or upstream regulators. Such experiments would certainly move this paper out of the realm of being entirely descriptive in nature.

Reviewer #3 (Remarks to the Author):

The manuscript "Chromatin organization in the female brain fluctuates across the oestrous cycle" presents a series of analyses comparing behavior, hormonal, neuronal, gene expression and chromatin structure among females in diestrus, estrus and males. As the authors indicate, the female brain is understudied in most contexts and this study identifies molecular changes associated with estrus state that may contribute to variability in the behavior and brain architecture of females and possibly contribute to sex differences in risk of psychopathology. The description of chromatin changes that accompany estrus state are a valuable contribution to the literature and identified candidate genes may be informative for future studies of sex differences and sex-hormone associated neural and behavioral changes.

Critiques:

1) It is unclear what phenomenon of risk serves as the framework for this study. On the behavioral measures, the focus on how diestrus females are different from proestrus females and males – so not strictly a sex-difference. On Lines 44 onward, the text describes major reproductive state

transitions rather than within-cycle transitions. Premenstrual dysphoric disorder seems to be a better fit but it is unclear whether the behavioral measures assessed capture that phenotype.

2) The study involves comparison on all measures between diestrus females, proestrus females and males. However, what is notably lacking from the analyses and not embedded in the design in the study is any possibility of looking at the correlation between different measures within individuals that would determine whether chromatin state is associated in any way with any other measure. Thus, no causal inference is possible and the authors can't suggest that they have identified functionally meaningful changes or that hormones are in fact the critical variable defining the changes identified. All measures may be completely unrelated to each other and driven by variables associated with state (DF, PF, M). Use of analytic or methodological strategies that could strengthen the causal inference would enhance the impactfulness of the work.

3) Conducting brain work in behaviorally tested mice may lead to gene expression changes associated with behavioral testing that would not otherwise be observed.

4) The difference in location of open chromatin in males and females (proestrus) is interesting – is the comparable region open in males not open in females or does it not pass correction for multiple hypothesis testing?

5) At several points in the manuscript there is a choice to pursue candidate genes from among many that would be of plausible interest. What is the basis of selection or rather of not selecting other plausible candidates?

6) Was the side of the brain used (left or right) for ventral hippocampus consistent across individuals?

7) Confirm that your endogenous reference gene used for gene expression analyses is not modulated by state.

Response to Reviewer's Comments

We would like to thank the reviewers for their time and effort as well as for thoughtful comments and suggestions which, we believe, have resulted in much improved manuscript. As suggested by the editor and reviewers, we have now performed RNA-seq analysis on sorted neuronal nuclei from ventral hippocampal tissue of proestrus, dioestrus, and male mice. We also integrated the ATAC-seq and RNA-seq results, revealing evidence supporting the functional significance of chromatin organizational changes across the estrous cycle and between sexes. These new analyses not only strengthen our original conclusions but also reveal new candidate genes and provide new insights into mechanisms underlying the oestrous cycle- and sex-dependent chromatin and behavioral variability. Our revised manuscript includes: two new figures in the main text (*Figures 4 and 5*); one supplementary figure (*Supplementary Figure 7*); six supplementary tables (*Supplementary Tables 6-10 and 12*); and about 2,500 words of the added text. Below please find a point-by-point response to all reviewers' comments. The text was also revised accordingly.

Reviewer #1

- We would like to thank the reviewer for the careful analysis, kind comments, and a very positive review of our study. Please find below a response to your comment.

There are a few experimental strategies that would be helpful additions, but to avoid acting as an additional author on the paper, these are only suggestions that I do believe are not necessary inclusions for the publication of this manuscript.

1. Given the emphasis on shifts in ER receptor activation, it would be of interest to demonstrate that membrane-bound ER is indeed the driver of these changes. There are several methods for achieving this, which include ovariectomy with and without E2 replacement. To differentiate between effects of membrane-bound and nuclear ER, it would be helpful to include BSA-bound E2 that cannot cross the membrane, thus limiting the effects to only membrane bound receptors.

- We appreciate this suggestion. We agree that going deeper into the receptor mechanism will be an important next step, but as an extension of this study and not as part of this manuscript. As the reviewer suggested, these experiments will require ovariectomy and E2 replacement. The major approach (and we believe also strength) of this study is that we performed all the analyses, including the new RNA-seq study, under physiological conditions, and were able to show extensive estrous cycle-dependent changes in chromatin organization, gene expression, and behavior. For this manuscript, we would like to keep this focus and, again, we thank the reviewer for the thoughtful comments and great suggestions for the future work.

Reviewer #2

- We thank the reviewer for the careful analysis and for thoughtful comments and suggestions which have significantly improved our manuscript. Please find below responses to all your comments.

1) As it stands, given its descriptive nature, this manuscript reads a lot like a Resource article, which is not necessarily a bad thing. However, if the authors choose to keep the manuscript descriptive, then they should consider expanding their analyses to include multiple brain regions in order to examine whether hormonal fluctuations result in similar or dissimilar outcomes brain-wide. To do so, the authors may elect to choose one other brain region associated with mood disorders (e.g., mPFC, ventral striatum, etc.), along with one that is not typically associated with the behaviors that they are assessing (e.g., visual cortex).

- Thanks to the reviewer for this comment. As suggested by the editor and reviewers, the resource value of the manuscript is now supplemented with the addition of RNA-seq analysis of sorted neuronal nuclei isolated from ventral hippocampal tissue of proestrus, dioestrus, and male mice. We have also integrated the ATAC-seq and RNA-seq data to strengthen our conclusions. Please see the added material: new Figures 4 and 5 and Supplementary Tables 6-10 and 12. The added text is also highlighted in the main manuscript text and consists of:

- New subsections in the Results section (Lines 253-394) titled as: “Nuclear RNA expression in ventral hippocampal neurons varies with the oestrous cycle stage and sex” and “Chromatin remodelling- and anxiety-relevant genes exhibit differential expression and differential chromatin accessibility across the oestrous cycle”.
- Added discussion (Lines 434-437 and 446-468).

2) While their candidate-based approach to testing whether genes displaying alterations in chromatin accessibility also associate with changes in expression is reasonable, it would be helpful to understand on a more global level whether such correlations exist genome-wide. Therefore, the authors should perform RNA-seq on sorted neurons across the oestrous cycle to determine this more accurately.

- This is a great suggestion. As requested by the reviewer, we have performed RNA-seq on sorted neuronal nuclei and have included these data in the manuscript and discussed them in light of our ATAC-seq findings (Please see Response to Reviewer’s critique #1 for the details of all added materials). We believe that the inclusion of the nuclear RNA-seq data has significantly improved our manuscript. These new data not only strengthen our conclusions but provide new candidate genes and novel insights into the mechanism underlying the observed chromatin and behavioral changes (Please see the newly added materials including Results and Discussion).

3) While the upstream motif analyses are interesting, this part of the story needs to be further developed. For example, if the authors believe that Egr1 is driving a transcriptional program associated with dioestrus, then they should perform manipulations (e.g., viral OE) of Egr1, followed by ATAC-seq (or targeted ATAC-PCR) to verify this effect directly.

- We recognize that the studies implicating Egr1, while novel, are not yet definitive. In this manuscript, we focus on the effects of natural hormonal shifts on chromatin, gene expression, and behavior. As of now, there is no available viral expression system that would allow for cycling changes in gene expression that would mimic the natural estrous cycle. The AAV system requires three weeks to achieve stable overexpression. Even the HSV viral system, which allows a more transient expression, would not allow for the controlled cycling expression that we need to re-

create natural conditions. In addition, we would need to perform an ovariectomy which, by itself, would certainly induce some adaptive changes in the brain, including changes in gene expression. Therefore, we do not believe that the current tools will allow us to mimic the system that we are studying, although this will be one of the most important questions that we are planning to address in our future work. However, with the added RNA-seq data, we show that the Egr1 motif is enriched in the genes that show concomitant differential expression and varying chromatin accessibility between proestrus and dioestrus. This also includes genes encoding chromatin remodeling factors, further supporting the role of Egr1 in the oestrous cycle-dependent chromatin and transcriptional regulation. Please see the new **Figure 5b-c**. We also added the following text in the Results section (Lines 355-360):

“Next, we explored whether the “overlapping genes”, showing both differential gene expression and differential chromatin accessibility between proestrus and dioestrus, share any transcription factor binding site within the range of ± 1 kb from the TSS. Remarkably, this analysis revealed that the Egr1 motif was one of the top transcription factor binding sites found in the regulatory regions of 21 (24.1%) overlapping genes, including all genes of the main “chromatin remodeling cluster” (Chd3, Chd4, Chd6, Smarcc2, and Ncor2) (Figure 5b).”

- This was also discussed in the Discussion section (Lines 434-437):

“We also show that the expression of ATP-dependent chromatin regulators, all putative targets of Egr1, is increased in proestrus, providing an additional insight into mechanisms driving chromatin re-organization across the oestrus cycle.”

4) The spine data in Fig. 3e, while interesting, seem out of place in this manuscript unless the authors wish to further link manipulations in chromatin accessibility to these changes. In other words, if they want to link these phenomena, then I would suggest performing manipulations of a specific ATAC targets or upstream regulators (perhaps Egr1) to show that this will alter the differences in spines observed throughout the oestrous cycle.

- We appreciate where the reviewer’s concerns lie, but would like to respectfully disagree that dendritic spine data are out of place in this manuscript. We are showing links between chromatin organizational changes and changes in both behavior and brain structure phenotypes. The enriched GO terms (**Figure 1f**) and KEGG pathways (**Figure 2a**) show differential chromatin accessibility among genes important for synaptic structure and function. Egr1 is known to be important for synaptic plasticity. The genes that we selected, also Egr1 target genes, *Ncan*, *Gria3*, and *Ptprr1* (**Figure 3d**), are all found to be important for dendritic spine density and synaptic plasticity. For instance, it has been shown that an overexpression of PTPRT increases the density of dendritic spines (**REF 41**). Therefore, we believe that dendritic spine data add to the manuscript and are well connected with the discussion of genes important for synaptic plasticity.

Similarly, it would be nice if the authors could link their behavioral analyses in Fig. 1 to alterations in accessibility/gene expression by manipulating targets and/or upstream regulators. Such experiments would certainly move this paper out of the realm of being entirely descriptive in nature.

- To address this concern, we added the RNA-seq analysis described earlier. With these added RNA-seq data, we are now providing new candidate genes differentially expressed and with

different chromatin accessibility between proestrus and dioestrus, including *Lamp5*, *Dlk1*, *Dkk1*, and *Chd3*. Importantly, these genes have been previously manipulated by other researchers, showing that their down-regulation or up-regulation induce anxiety- or depression-like behavior. We provide a detailed discussion of each of these genes in light of the published studies and in connection to our data, further providing the link between behavioral variability and chromatin accessibility/gene expression changes across the oestrous cycle.

Please see the added text in the Results section:

Lines (301-320): “To explore candidate genes of relevance to anxiety behaviour and chromatin organization, we selected some of the top genes from the proestrus-dioestrus comparison: *Lamp5* (encoding Lysosome-Associated Membrane Protein 5), *Dkk1* (encoding Dickkopf Like Acrosomal Protein 1), *Chd3* (encoding Chromodomain Helicase DNA Binding Protein 3), and *Smarcc2* (encoding SWI/SNF Related, Matrix Associated, Actin Dependent Regulator Of Chromatin Subfamily C Member 2), and validated their differential nuclear RNA expression in ventral hippocampal neurons of proestrus and dioestrus females (**Figure 4c**). *Lamp5* is a brain-specific LAMP family member, specifically implicated in trafficking and sorting of synaptic proteins as well as in synaptic plasticity in GABAergic neurons^{48,49}. We found this gene to be of particular interest because *Lamp5* mutant male mice show decreased anxiety-like behavior compared to their wild type counterparts⁴⁸. In our study, we found that *Lamp5* shows lower expression in proestrus than in dioestrus females (**Figure 4c**), which is consistent with lower anxiety levels in proestrus compared to dioestrus (**Figure 1b**) and with the behavioural phenotype of the *Lamp5* *-/-* mice⁴⁸. Another gene of interest was *Dkk1*, which was previously identified as a hub gene in the ventral hippocampus, regulating the activity of a gene network implicated in stress susceptibility⁵⁰. Of note was that the overexpression of *Dkk1* in the ventral hippocampus, our area of interest, has been shown to induce increased depression-like behaviour in response to social defeat stress in male mice⁵⁰. In our study we show that this gene is up-regulated in dioestrus compared to proestrus females (**Figure 4c**), suggesting a possible role of cycling *Dkk1* expression in regulating anxiety levels across the oestrus cycle.”

Lines (321-329): “Both *Chd3* and *Smarcc2* encode ATP-dependent chromatin remodelling factors which regulate chromatin compaction and the accessibility of nucleosomal DNA through a catalytic process powered by ATP hydrolysis⁵¹. *Chd3* is an integral subunit of the Mi-2/NuRD chromatin-remodeling histone deacetylase complex⁵². Interestingly, a recent study has compared *Chd3* protein levels in the ventral hippocampus of male mice selectively bred for either high anxiety behavior or normal anxiety-related behavior, showing that lower *Chd3* levels are associated with high anxiety phenotype⁵³. In line with this, we found that *Chd3* is up-regulated in proestrus compared to dioestrus females (**Figure 4c**), implying that changing *Chd3* levels may contribute to both chromatin reorganization and variation in anxiety behavior during the oestrous cycle.”

Lines (372-387): “It is also plausible that chromatin closing and down-regulation of *Dlk1* expression in ventral hippocampal neurons contributes to the high anxiety phenotype observed during dioestrus (**Figures 1b and 5c**). *Dlk1* is an imprinted, paternally-expressed gene, which is epigenetically regulated and strongly implicated in embryonic development⁵⁴. Only recent studies have demonstrated the role of *Dlk1* in postnatal neurogenesis⁵⁴, and we know that *Dlk1* expression in the adult brain is very restricted, largely to the reward system^{54,55}. However, this gene has been recently identified as a specific transcriptional marker of the ventral hippocampus⁵⁶, with no detectable expression in other hippocampal areas, suggesting that *Dlk1*

may have a specific role in the regulation of emotion and reward. Consistent with this, a recent study has shown that *Dlk1* deletion in male mice is associated with increased anxiety-like behaviors in the elevated plus maze, open-field, and light-dark box tests⁵⁵. Remarkably, dioestrus females show high anxiety in all these tests (**Figure 1b**) and have significantly down-regulated *Dlk1* expression compared to proestrus females (**Figure 5c**). Notably, these findings indicate that neuronal expression of an imprinted gene is dynamically regulated in the adult female brain, via chromatin re-organization, likely contributing to varying anxiety levels across the oestrous cycle.“

Reviewer #3

- Again, we thank this reviewer for their careful analysis of our manuscript and for raising some important points that required clarification. We have included additional text in the manuscript to address those critiques and are also providing the detailed response to each critique.

1) It is unclear what phenomenon of risk serves as the framework for this study. On the behavioral measures, the focus on how diestrus females are different from proestrus females and males – so not strictly a sex-difference. On Lines 44 onward, the text describes major reproductive state transitions rather than within-cycle transitions. Premenstrual dysphoric disorder seems to be a better fit but it is unclear whether the behavioral measures assessed capture that phenotype.

- This was a valuable question for us, as it indicated that the framework of our study requires clarification. Our study is physiological and is not a typical model of anxiety or depression which would require an environmental stressor and/or genetic manipulation. Rather, we focus on and “model” the increased female vulnerability to anxiety/depression due to natural hormonal shifts. To clarify the framework of our study, we include a new text providing further details on: a) clinical evidence that natural hormone fluctuations and, in particular, a physiological drop in estrogen increases risk for anxiety/depression; b) how our study models this increased vulnerability; c) how results of our study provide an insight into mechanism(s) underlying increased female vulnerability to these disorders (Lines 446-466).

“From human studies, we know that the increased female risk for these disorders is strongly associated with hormonal fluctuations, as evidenced by the jump in risk coinciding with the onset of menarche and perimenopause and during the post-partum period. A subset of women also suffer from premenstrual dysphoric disorder and, strikingly, women with major depression, although under antidepressant treatment, often experience worsening of their symptoms in the low-estrogenic, premenstrual phase of the cycle^{64,65}. Here we provide a mechanistic insight into how this female vulnerability may be mediated. We reproduce sex- and oestrous cycle-dependent anxiety behaviour in our mouse model, showing that a physiological drop in estrogen in females results in higher anxiety levels, both compared to their high-estrogenic phase and to males. We further show that the oestrous cycle stage significantly affects chromatin organization and expression of genes relevant to serotonergic transmission and anxiety behaviour. It has been known that fluctuating oestrogen levels affect serotonergic function³¹ but here we provide an epigenetic transcriptional mechanism contributing to these changes. In addition, it is particularly interesting to note that our candidate genes revealed by the nucRNA-seq analysis such as *Lamp5*, *Dkk1*, *Chd3*, and *Dlk1* are shown to confer vulnerability to anxiety- and depression-related behaviours in males under extreme conditions such as genetic deletion or overexpression, chronic stress exposure, or selective breeding^{48,50,53,55}. However, the expression of these same

genes naturally cycles in females, reaching behavioural risk-associated levels at the time when endogenous oestrogen levels drop, and likely contributing to inherent, female-specific vulnerability to depression and anxiety disorders.”

2) The study involves comparison on all measures between diestrus females, proestrus females and males. However, what is notably lacking from the analyses and not embedded in the design in the study is any possibility of looking at the correlation between different measures within individuals that would determine whether chromatin state is associated in any way with any other measure. Thus, no causal inference is possible and the authors can't suggest that they have identified functionally meaningful changes or that hormones are in fact the critical variable defining the changes identified. All measures may be completely unrelated to each other and driven by variables associated with state (DF, PF, M). Use of analytic or methodological strategies that could strengthen the causal inference would enhance the impactfulness of the work.

- The reviewer describes an ideal scenario, but in practice we are not able to run the correlation between different measures within individuals due to the fact that behavioral testing *per se* can impact epigenetic modifications and gene expression, as pointed out by the reviewer in critique #3 (below). In this manuscript, we focus on natural hormonal shifts and do not perform any manipulations that could, by themselves, induce many non-physiological changes in chromatin and gene expression. However, we do provide strong links between our data and published studies that implicated our candidate genes in anxiety- and depression related phenotypes. To strengthen our conclusions, we now also provide the new RNA-seq data and new candidate genes, and have significantly expanded our Results and Discussion sections. Please see the new Figures 4 and 5, new Supplementary Tables 6-10 and 12, and the added text in the Results (Lines 253-394) and Discussion (Lines 434-437 and 446-468) sections.

3) Conducting brain work in behaviorally tested mice may lead to gene expression changes associated with behavioral testing that would not otherwise be observed.

- We completely agree with the reviewer and this is exactly the reason why we performed our ATAC-seq and nucRNA-seq experiments in animals that were not behaviorally tested. However, this is also the reason why we are unable to do the correlation analysis of molecular and behavioral outcomes in the same animals, as explained in our answer to the reviewer's critique #2.

4) The difference in location of open chromatin in males and females (proestrus) is interesting – is the comparable region open in males not open in females or does it not pass correction for multiple hypothesis testing?

- Thanks to the reviewer for highlighting this important finding – the varying chromatin state as a function of the estrous cycle stage and sex is not only “gene-specific” but can also be “location-specific” within the same gene. We give an example of this, the upstream region of the *Ppp1r1b* gene. To answer the reviewer's question, the “male peak” was only present in males and was not found in proestrus or dioestrus females. Technically speaking, this means that a MACS2-

called “peak” was detected in at least two animals of the male group but not in proestrus or dioestrus females. We analyzed the data in this way and called peaks at $q < 0.05$ so that only high confidence peaks are considered (please see the methods section, Lines 632-637). While we give only one example of this “location-specific” peak in our manuscript due to space limitation, we have provided all differential peaks in our Supplementary Table 1 so that other researchers can look at their genes of interest and find sex- and estrous cycle-specific regions.

5) At several points in the manuscript there is a choice to pursue candidate genes from among many that would be of plausible interest. What is the basis of selection or rather of not selecting other plausible candidates?

- In the original version of the manuscript, we selected three groups of genes: 1) Genes generally important for neuronal function – *Ppp1r1b* (a gene encoding Darpp32), *Kcnv1* (gene encoding a potassium channel), and *Syn1* (a gene encoding a synaptic vesicle protein); 2) Genes which are part of serotonergic synapse pathway and have been implicated in anxiety/depression (*Htr2b*, *Cacna1c*, *Ptgs1*); and 3) Genes that are *Egr1* targets and are important for synaptic plasticity (*Ncan*, *Gria3*, and *Ptprt*). With these examples, we wanted to show that differential chromatin accessibility is found in genes important for various neuronal functions and anxiety behavior. Particular genes from each of these groups were selected based on the position of differential ATAC peak(s) – we selected peaks occurring in the likely regulatory regions, close and upstream of the TSS. However, these were only examples and peaks occurring in other genomic regions can have equally important roles in gene regulation and, for this reason, we provide all differential ATAC-seq peaks with the associated genes in all comparisons (**Suppl Table 1**), so that other researchers can find and test their genes and genomic regions of interest.

- With the new nucRNA-seq analysis, we provide lists of all differentially expressed genes in all comparisons (**Suppl Table 6**). We then focus our analysis and discussion on some of the top significant genes which show differential chromatin accessibility between proestrus and dioestrus and which have been implicated in anxiety behavior and chromatin organization. Please see the new Figures 4 and 5, new Supplementary Tables 6-10 and 12, and the added text in the Results (Lines 253-394) and Discussion (Lines 434-437 and 446-468) sections.

6) Was the side of the brain used (left or right) for ventral hippocampus consistent across individuals?

- We agree with the reviewer that this point requires clarification. For the ATAC-seq experiments, the side of the brain was randomly selected in each animal/group so that our data are not biased toward one side of the brain. We clarified this in the methods section which now reads as follows (Lines 573-575): *“the ventral hippocampus was randomly dissected from one side of the brain from each animal (equally representing the left and right hippocampi), then frozen in liquid nitrogen, and later processed.”*

For the newly added nucRNA-seq data, we did bilateral ventral hippocampal dissections and pooled the tissue from two animals for each sample, as this method required a larger number of nuclei compared to ATAC-seq. This is also clearly stated in the methods section (Lines 683-685): *“As this assay requires a larger amount of starting material, the ventral hippocampus was*

dissected bilaterally from each animal and brain tissue from two animals was pooled for each biological replicate (n=3 replicates/group)."

7) Confirm that your endogenous reference gene used for gene expression analyses is not modulated by state.

- This is an important comment which we appreciate. *Ppia* was selected as a reference gene because its expression is not affected by the estrous cycle stage or sex. We have now shown this for both whole-cell and nuclear RNA expression and have included this statement in the methods section of the paper (Lines 676-678).

"Ppia was used as a reference gene for both whole-cell and nuclear RNA expression analysis, as we have confirmed that, in either case, the expression of this gene does not vary with the oestrous cycle stage or sex."

REVIEWERS' COMMENTS:

Reviewer #2 (Remarks to the Author):

The authors have done a very nice job in responding to my previous critiques. The inclusion of nucRNA-seq adds considerably to the manuscript and supports their ATAC-seq findings. I do still feel strongly that this paper reads more like a Resource article (owing to a lack of functional manipulations/validations of gene expression/chromatin accessibility differences between hormonal states), but I think that this is okay. The manuscript addresses an important, and very much understudied, component of brain function (i.e., the impact of hormonal cycling on chromatin organization in the brain), and I feel that such a resource will be important to the field. Thus, I feel that the manuscript is now suitable for publication in Nature Communications.

Reviewer #3 (Remarks to the Author):

Though descriptive in nature, the manuscript provides a detailed analyses of the molecular changes that correlate with oestrous state and could form the basis of more definitive studies of the impact of hormones on molecular changes in the brain. The authors have been responsive to previous critiques.

RESPONSE TO REVIEWERS' COMMENTS

- We would like to thank the reviewers for their time and effort as well as for their very positive reviews of our revised manuscript. The reviewers did not request any additional changes to the manuscript (please see below).

Reviewer #2 (Remarks to the Author):

The authors have done a very nice job in responding to my previous critiques. The inclusion of nucRNA-seq adds considerably to the manuscript and supports their ATAC-seq findings. I do still feel strongly that this paper reads more like a Resource article (owing to a lack of functional manipulations/validations of gene expression/chromatin accessibility differences between hormonal states), but I think that this is okay. The manuscript addresses an important, and very much understudied, component of brain function (i.e., the impact of hormonal cycling on chromatin organization in the brain), and I feel that such a resource will be important to the field. Thus, I feel that the manuscript is now suitable for publication in Nature Communications.

Reviewer #3 (Remarks to the Author):

Though descriptive in nature, the manuscript provides a detailed analyses of the molecular changes that correlate with oestrous state and could form the basis of more definitive studies of the impact of hormones on molecular changes in the brain. The authors have been responsive to previous critiques.